



# Representation of the Denmark Strait Overflow in a z-coordinate eddying configuration of the NEMO (v3.6) ocean model: Resolution and parameter impacts

Pedro Colombo[1], Bernard Barnier[1,4], Thierry Penduff[1], Jérôme Chanut[2], Julie Deshayes[3], Jean-Marc Molines[1], Julien Le Sommer[1], Polina Verezemskaya[4], Sergey Gulev[4], and Anne-Marie Treguier[5]

[1]Institut des Géosciences de l'Environnement, CNRS-UGA, Grenoble, 38050, France
[2]Mercator Ocean International, Ramonville Saint-Agne, France
[3]Sorbonne Universités (UPMC, Univ Paris 06)-CNRS-IRD-MNHN, LOCEAN Laboratory, Paris, France
[4]P. P. Shirshov Institute of Oceanology, Russian Academy of Sciences, Moscow, Russia
[5]Laboratoire d'Océanographie Physique et Spatiale, Brest, France

**Correspondence:** Pedro Colombo (pedro.colombo@univ-grenoble-alpes.fr)

**Abstract.** We investigate in this paper the sensitivity of the representation of the Denmark Strait overflow produced by a regional z-coordinate configuration of NEMO (version 3.6) to the horizontal and vertical grid resolutions and to various numerical and physical parameters. Three different horizontal resolutions, 1/12°, 1/36° and 1/60°, are used respectively with 46, 75, 150 and 300 vertical levels. Contrary to expectations, in the given numerical set-up, the increase of the vertical resolution

did not bring improvement at eddy-permitting resolution (1/12°). We find a greater dilution of the overflow as the number of vertical level increases, and the worse solution is the one with 300 vertical levels. It is found that when the local slope of the grid is weaker than the slope of the topography the result is a more diluted vein. Such a grid enhances the dilution of the plume in the ambient fluid and produces its thickening. Although the greater number of levels allows for a better resolution of the ageostrophic Ekman flow in the bottom layer, the final result also depends on how the local grid slope matches the topo-

graphic slope. We also find that for a fixed number of levels, the representation of the overflow is improved when horizontal resolution is increased to 1/36° and 1/60°, the most drastic improvements being obtained with 150 levels. With such number of vertical levels, the enhanced vertical mixing associated with the step-like representation of the topography remains limited to a thin bottom layer representing a minor portion of the overflow. Two major additional players contribute to the sinking of the overflow, the breaking of the overflow into boluses of dense water which contribute to spread the overflow waters along the

Greenland shelf and within the Irminger Basin, and the resolved vertical shear that results from the resolution of the bottom Ekman boundary layer dynamics. This improves the accuracy of the calculation of the entrainment by the turbulent kinetic energy mixing scheme (as it depends on the local shear), and improves the properties of the overflow waters such that they more favorably compare with observations. At 300 vertical levels the dilution is again increased for all horizontal resolutions. The impact on the overflow representation of many other numerical parameters were tested (momentum advection scheme,

lateral friction, bottom boundary layer parameterisation, closure parameterisation, etc.) but none had a significant impact on the overflow representation.





# 1 Introduction

An oceanic overflow is a dense water mass formed on a continental shelf or in a marginal sea that joins the open ocean by overflowing topographic constraints like narrow straits or sills, or by flowing down steep topographic slopes. Initially pushed by its greater density towards the interior of the ocean it is structured as a plume or boluses of dense fluid, cascading down and

mixing with ambient waters and finally flowing along topographic slopes at a depth close to a neutral buoyancy level (Legg et al. (2009)). Overflows of important magnitude are those associated with the straits of the Nordic Seas (i.e. the Denmark Strait Girton and Standford (2003), Brearley et al. (2012) and the Faroe Bank Channel Hansen and Østerhus (2007)), of the Mediterranean Sea (Baringer and Price (1997)), the Red Sea (Peters et al. (2005)), and the continental shelves of the polar oceans (Killworth (1977), Baines and Condie (1998)), see Legg et al. (2009), Magaldi et al. (2015), Mastropole et al. (2017),

for more reference papers). Altogether, these overflows feed most of the world ocean deep waters. This flux of dense water to the deep ocean is balanced by the intrusion of waters from regions different from where the overflow waters are formed. For example, the flux of cold waters formed in the Arctic Ocean and the Nordic Seas that enters the North Atlantic with the Denmark Strait and Faroe Bank Channel overflows is balanced by warm and salty Atlantic waters that flows over the Iceland-Scotland Ridge towards the Arctic Ocean via upper ocean currents (Hansen and Østerhus (2007)). Consequently, overflows

play an important role distributing heat and salt in the ocean. For the case of the Denmark Strait overflow (DSO hereafter), it feeds the Deep Western Boundary Current in the North Atlantic, and so contributes to the global thermohaline circulation (Dickson and Brown (1994), Beismann and Barnier (2004), Dickson et al. (2008), Yashayaev and Dickson (2008), Danabasoglu et al. (2010), Zhang et al. (2011), von Appen et al. (2014)). This world-ocean wide importance of the overflows makes their representation a key aspect of ocean general circulation models (OGCMs).

However, the dynamical processes that control overflows have rather small scales and are not resolved in present large-scale coarse resolution (non-eddying) ocean models used for climate studies (Legg et al. (2008)). It is also known that models using fixed geopotential levels as vertical coordinate (i.e. z-level models) generate spurious vertical mixing at that resolution when moving dense overflow waters downslope. The link of this spurious mixing with the staircase-like representation of the bottom topography peculiar to these models is well established (Winton et al. (1998), Wang et al. (2008)). The parameterisation of

overflows in these models has been the topic of a number of studies (Beckmann and Döscher (1997), Campin et al. (2012), Killworth and Edwards (1999), Song and Chao (2000), Danabasoglu et al. (2010), Wang et al. (2015)). A large number of idealized model studies, many of them conducted in the DOME framework (Dynamics of Overflows Mixing and Entrainment, Legg et al. (2006), Legg et al. (2009)), tested the ability of overflow parameterizations against very high-resolution simulations in a variety of OGCMs. When used in global simulations these parameterisations improve overflows, but still produce

deep/bottom water properties that are not yet satisfactory if not inadequate (Condie et al. (1995); Griffies et al. (2000); Legg et al. (2009); Danabasoglu et al. (2010), Danabasoglu et al. (2014), Downes et al. (2011); Weijer (2012); Heuzé et al. (2013), Wang et al. (2015), Snow et al. (2015)). Past model studies performed with DOME-like idealized configurations also permitted to gain understanding on the dynamics of overflows and on the sensitivity of their representation in models to physical and numerical parameters (see Reckinger et al. (2015) for exhaustive references and a synthesis of the main findings). Significant



differences between models due to the type of vertical coordinate system were pointed out (e.g. Ezer and Mellor (2004), Legg et al. (2006), Wang et al. (2008), Laanaia et al. (2010), Wobus et al. (2011), Reckinger et al. (2015)).

Numerical modelling of dense water cascades with OGCMs designed to simulate the large scale circulation still represents a challenge. A first complication arises from the hydrostatic approximation used in the derivation of the vertical momentum

equation which yields to neglect the vertical acceleration terms and to misrepresent the vertical mixing processes (Özgökmen (2004)). A second complication arises from the grid-resolution of current global eddying models used for multi-decadal hindcast simulations, which at the latitude of the Nordic Seas (e.g. $60°N$) is typically of the order of 5 km to 12 km (i.e. $1/12°$ to $1/4°$, DRAKKAR Group (2014)). Magaldi et al. (2015), compared high-resolution (2 km) hydrostatic and non-hydrostatic simulations of dense water cascading in a realistic model configuration of the Irminger basin. They found that for such  2

km horizontal resolution, the parameterization of the non-resolved turbulence used in the hydrostatic model was accurately representing the effects of the lateral stirring and vertical mixing associated with the cascading process.

Meanwhile most recent high-resolution regional modelling studies of the Denmark Strait overflow (Magaldi et al. (2011), Magaldi et al. (2015), Koszalka et al. (2013), Koszalka et al. (2017)) or the Faroe Bank Chanel overflow (Riemenschneider and Legg (2007), Seim et al. (2010)) have been using hydrostatic model configurations of the MIT gcm. These studies, as

they provide modelled overflows in good agreement with observations, significantly improved the actual understanding of the overflows and their modelling. For the case of the DSO, the studies referred above especially pointed out the importance of the resolution of the cyclonic eddies linked to the dense overflow water boluses on the entrainment, and the importance of the dense water cascading from the East Greenland Shelf with the Spill Jet. On the modelling aspects, these studies provided some rationale regarding the grid-resolution that permit a representation of the overflows that agrees with observations (a resolution

of 2 km in the horizontal and a few tens of meters near the bottom in the vertical). They also characterized the dependence on various model parameters regarding the mixing of the overflow waters with ambient waters. For the case of the Faroe Bank Channel overflow for example, Riemenschneider and Legg (2007) found a greater sensitivity of the mixing to horizontal resolution and, but to a lesser extent, to vertical resolution and vertical viscosity. The sensitivity to other parameters tested (bottom drag coefficient, strength of the inflow) were found to be minor. However, the high resolution used in these regional

studies cannot yet be used in eddying global model hindcast simulations of the last few decades or for eddying ensemble simulations.

Global OGCM are now commonly used at resolutions of $1/12°$, which yields a grid-size of about 5 km in the region of the Nordic Seas overflows and may resolve with some accuracy the entrainment of ambient waters into the overflow plume by eddy-driven advection, but not the small-scale diapycnal mixing which still needs to be fully parameterized by the turbulent

closure scheme. Chang et al. (2009), studied the influence of horizontal resolution on the relative magnitudes and pathways of the Denmark Strait and Iceland-Scotland overflows in a North Atlantic configuration of the HYCOM OGCM (Chassignet et al. (2003)). They found that at $1/12°$, the highest resolution tested, the simulations show realistic overflow transports and pathways and reasonable North Atlantic three-dimensional temperature and salinity fields. The ability of HYCOM to represent the spreading of the overflow waters at $1/12°$ resolution was later confirmed by the studies of Xu et al. (2010), Xu et al.

(2014). Marzocchi et al. (2015), provided an assessment of the ocean circulation in the subpolar North Atlantic in a 30-years





long hindcast simulation performed with the ORCA12 configuration, a z-coordinate partial-step global implementation of the NEMO OGCM (Madec et al. (2016)) at $1/12°$ resolution developed by the Drakkar Group (DRAKKAR Group (2014)). They found that the model had some skills as the volume transport and variability of the overflows from the Nordic Seas were reasonably well represented. However, significant flaws were found in the overflow water mass properties that were too warm

and salty. This latter bias can be partly attributed to the excessive entrainment peculiar to the z-coordinate, but other sources of biases, like the warm and salty bias found in the entrained waters of the Irminger basin, a resisting bias in this type of model simulations (Treguier (2005), Rattan et al. (2010)), are likely to contribute.

Despite the progresses reported above, it is clear that overflow representation is still a resisting flaw in z-coordinate hydrostatic ocean models. NEMO (version 3.6) is now commonly used in eddying ($1/4°$ to $1/12°$) configurations for global

or basin-scale, climate-oriented studies (e.g. Megan et al. (2014), Williams et al. (2015), Treguier et al. (2017), Sérazin et al. (2018)), reanalyses and operational forecasts (Lellouche et al. (2013), Lellouche et al. (2018), Le Traon et al. (2017)), or ensemble multi-decadal hindcast simulations (Bessières et al. (2017), Penduff et al. (2018)). But despite their use by a growing community, model configurations like ORCA12 remain computationally expensive and sensitivity studies are limited. Therefore, there is a need to establish the sensitivity of the simulated overflows to the available parameterizations in a realistic

framework relevant to the commonly used resolutions.

The objective of this work is to provide a comprehensive assessment of the representation of overflows by NEMO in a realistic eddy-permitting to eddy-resolving configuration that is relevant for many present global simulations performed with this model, in particular with the standard $1/12°$ ORCA12 configuration setup similar to that presently used for operational forecasting by the CMEMS[1]. Therefore, we limit our investigation to the sensitivity of the overflow representation when

standard parameters or resolution are varied, the objective being to identify the model parameters and resolutions of significant influence. However, because NEMO is also used at much higher resolution ($1/60°$, e.g. Ducousso et al. (2017.)) and offers possibilities of local grid refinement (Debreu et al. (2007)) already used with success (e.g. Chanut et al. (2008), Biastoch et al. (2009)), the use of a local grid refinement in overflow regions is also investigated. The approach is to set-up a regional model configuration that includes an overflow region that is similar, in terms of resolution and physical or numerical parameters, to

the global ocean eddying configurations widely used in the NEMO community. The DSO is chosen as test case because of its great magnitude and the relatively large amount of observations available. Considering that mesoscale eddies are not fully resolved at this resolution, the focus is on the overflow mean product and not on the details of the dynamics as it is done in the very-high resolution ( 2 km) studies of Magaldi et al. (2015) and Koszalka et al. (2017).

This work is presented in 3 parts. In the first part (Section 2), we present the method used to carry out the sensitivity tests.

We first describe the regional NEMO z-coordinate configuration developed to simulate the DSO, and the initial and forcing conditions common to all sensitivity simulations. The simulation strategy and the diagnostics developed for the assessment of the model sensitivity are described. The control simulation that represents a standard solution is run and diagnosed. In the second part (Section 3), we describe the sensitivity of the modelled overflow to a large number of parameters. Results from about 50 simulations are used, spanning vertical resolution (46, 75, 150, and 300 vertical levels), horizontal resolution ($1/12°$,

---

[1]Copernicus Marine Environment Monitoring Services: http://marine.copernicus.eu/





1/36° and 1/60°), lateral boundary condition (free slip and no-slip), bottom boundary layer parameterization, closure scheme, momentum advection scheme, etc. In the third part (Section 4), we describe in details the DSO produced by our best solution. We conclude the study with a summary of the main findings and some perspectives to this work.

## 2 Methods

### 2.1 Reference regional model configuration

We briefly describe the regional model configuration of reference used for the control run (changes being made afterwards in the different sensitivity tests). Version 3.6 of NEMO is used. The geographical domain is shown in Fig. 1. It includes part of the Greenland Sea, the Denmark Strait and a large part of the Irminger Sea. The reference NEMO setting has been designed to be representative of the solution that a global model would produce. Therefore, the configuration (geometry, numerical grid and schemes, physical parameterizations) has been extracted from an existing global ORCA12 configuration ($1/12°$ resolution, 46 z-levels) used in many simulations of the Drakkar Group (see Molines et al. (2014) for description and namelist). This configuration, referred to as DSO12.L46 (for $1/12°$ and 46 vertical levels) hereafter, is described with emphasis being given to parameters chosen for the control simulation from which sensitivity tests are performed. Changes that are made in the sensitivity tests are also indicated.

– Bottom topography and coastlines are exactly those of the global $1/12°$ ORCA12 configuration and are not changed in sensitivity experiments, except when grid refinement is used. In this latter case the refined topography is a bi-linear interpolation of that at $1/12°$, so the topographic slopes remain unchanged.

– The horizontal grid in the control run is a subset of the global tripolar grid at 1/12° (the so called ORCA12, $\sim 5km$ at the latitudes of the Irminger basin). The sensitivity to horizontal resolution is addressed by increasing the grid resolution to $1/36°$ ($\sim 2km$) and $1/60°$ ($\sim 1km$) over a small region that includes the Denmark Strait and a large part of the east Greenland shelf break (Fig. 1). The AGRIF 2-way grid refinement software (Debreu et al. (2007)) is used to connect the nested grids.

– Vertical resolution: The standard 46 fixed z-levels used in many Drakkar simulations are used in the Control simulation, with partial-steps to adjust the thickness of the bottom level to the true ocean depth Barnier et al. (2006). Sensitivity experiments also use 75, 150 and 300 z-levels. The cell-thickness as a function of depth is shown in Fig. A1.

– Momentum advection scheme: A vector invariant form of the momentum advection scheme (the energy and enstrophy conserving EEN scheme, Sadourny et al. (1975), Barnier et al. (2006) with the correction proposed in Ducousso et al. (2017.)) is used in the control and sensitivity experiments with an explicit biharmonic viscosity. Sensitivity tests used the upstream-biased third order scheme (UBS scheme) available in NEMO. Since this scheme includes a built-in biharmonic-like viscosity term with an eddy coefficient proportional to the velocity, no explicit viscosity is therefore used in the momentum equation when used.





- Isopycnal diffusivity on tracers: The TVD (Total Variance Diminishing) scheme standard in NEMO is used with the Laplacian diffusive operator rotated along isopycnal surfaces. The slope of the isopycnal surfaces are calculated with the standard NEMO algorithm. The diffusion coefficient remains the same in all sensitivity experiments. A sensitivity experiment was made that calculates the slope of the isopycnal using the Griffies Triad Algorithm Griffies (1998).

- Vertical mixing: it is treated with the standard NEMO TKE scheme Madec et al. (2016) Reffray et al. (2015). Because the model uses a hydrostatic pressure, the case of unstable stratification is treated with an Enhanced Vertical Diffusivity (EVD) scheme that sets the value of the vertical diffusion coefficient to $10m^2s^{-1}$ in case of static instability of the water column. It is applied on tracers and momentum to represent the mixing induced by the sinking of the dense water. A few sensitivity experiments used the EVD scheme on tracers only. Other experiments used the $K - \epsilon$ closure scheme
proposed in NEMO Reffray et al. (2015).

- Bottom boundary layer parameterization BBL: the control run does not use the BBL scheme that is available in NEMO, based on the parameterization of Beckmann and Döscher (1997). The scheme is tested in a sensitivity experiment.

- The free surface (linear filtered) scheme, the LIM2 sea-ice parameters, the scheme and data used at the lateral open boundaries, and the bulk formula and atmospheric forcing data that drive the model are identical in all experiments.

## 2.2   Initial conditions, surface and open boundary forcing

Data used to initialize the simulation and to drive the flow at the prescribed open boundaries are obtained from an ORCA12 simulation. This global simulation was initialized with temperature and salinity values from a climatology (Levitus, 1998) and started from rest. The atmospheric forcing that was used is the daily mean climatology of the 6-hourly DFS4.4 atmospheric forcing Brodeau et al. (2010). The forcing data of each day of the year is the climatological mean of that day calculated over
the period 1958 to 2001 (see Penduff et al. (2018), for details). This global simulation was run for almost 9 decades with this climatological forcing being repeated every year. It has also been used in the studies of e.g. Sérazin et al. (2015), or Grégorio et al. (2015) to study the intrinsic inter-annual variability. Every DSO model simulation used in the present study (the control run and all sensitivity runs) is initialized with the state of the global run on January $1^{st}$ of year 72 and is run for a period of 5 years (until year 76). The atmospheric forcing is the same as in the global run and it is also repeated every year. The data used
at the open boundaries of the DSO domain are extracted from years 72 to 76 of the global simulation (5 days mean outputs), so the open boundary forcing is fully consistent with the atmospheric forcing and the initial state.

We have chosen such a simulation scenario because several decades have passed from the initialization of the global run, and the model has reached a dynamical equilibrium and is close to thermodynamical equilibrium, which results in a negligible drift in the mass field. This allows to undoubtedly attribute the changes seen in the sensitivity experiments to the changes made
in the model setting. During this period the transport at the sill of the Denmark Strait is very stable and close to observed values ($\sim 3Sv$, see Section 2.3).



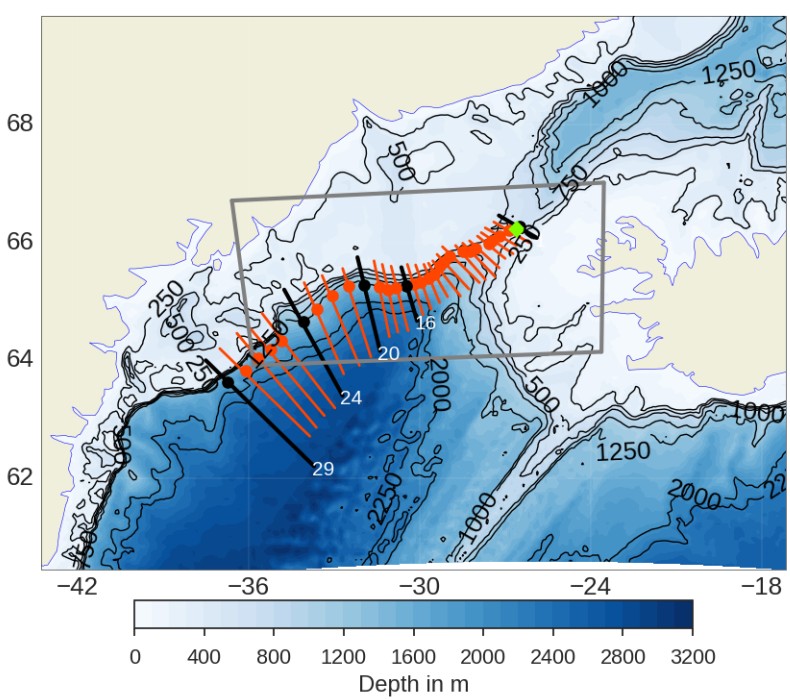

**Figure 1.** *Regional model domain. In colour the ocean depth. The 250, 500, 1000, 1500 and 2000 meter depth isobaths are contoured in black. The grey box indicates the region where the 2-way grid refinement (1/36° and 1/60°) is applied in some simulations. The location of the various sections used to monitor the model solution are shown by the red lines, and the numbered black lines for the most relevant ones. Section 1 is the reference section chosen for the sill. The thick dotted points indicate the center of the vein of the DSO in the Control simulation as calculated with the formulas given in Appendix B. The green dot indicates the reference point at the sill in order to calculate distances in Fig. 16.*

In this scenario, the initial bottom stratification is expected to be particularly affected by biases introduced in the properties of the water masses by the unrealistic representation of the overflows in the z-coordinate framework, so any improvement achieved in the representation of the overflow should be rapidly identified. However, the presence of these model biases reduces the relevance of the comparison to observations.

## 2.3 Control simulation DSO12.L46

A control simulation (referred to as DSO12.L46), is performed with the characteristics described in Section 2.1 and the initial and forcing conditions described above. As expected from its design, the solution of the DSO12.L46 5-year long control run reproduces very faithfully the solution of years 72 to 76 of the global ORCA12 run. This was verified in different aspects of the circulation. The large-scale circulation patterns is found to be very similar in both simulations, as illustrated with the





surface and bottom currents shown in Fig. 2. The pattern and amplitude of the predominant currents such as the East Greenland Current (EGC), the Irminger Current (IC) and the DSO itself are comparable in their smallest details. This circulation scheme also compares well with the circulation scheme described in Daniault et al. (2016) and the ORCA12 circulation described in Marzocchi et al. (2015). The correspondence between Global and Control runs regarding the properties of abyssal waters was

confirmed, especially at the 29 different sections along the path of the overflow as well as the correspondence in transport and bottom mean temperature across these sections (not shown). The bottom temperature in the Irminger basin was found to be very similar in both simulations, with a diluted signature of the overflow waters as expected from a z-level model after a simulation of several decades. Therefore this regional model appears as a reliable simulator of what the global model produces in that region.

Most important for the present study are the properties of the overflow "source waters", i.e. the properties of the waters at the sill of the Denmark Strait. Fig. 3 shows the volume transport at the sill of waters flowing below the 27.8 isopycnal. Both the global and the control runs show very similar mean and variability and a transport that is very steady during the 5 years of simulation. The model mean ($\sim 3Sv.$) is comparable to but in the lower range of the the values published in the work of Macrander et al. (2005) or Jochumsen et al. (2012). The standard deviation computed from 5-day outputs ($\sim 0.3Sv$ in the

control run) is rather small when compared to the $1.6Sv$ of Macrander et al. (2005).

Fig. 4b presents the characteristics of the mean flow across the sill. Compared with observations (see Fig. 1a of Macrander et al. (2007)), it shows a similar distribution of the isopycnals, specially the location of the 27.8 isopycnal. However, Mastropole et al. (2017) report waters denser than $28.0$ in the deepest part of the sill which the model does not reproduce. The distribution of velocities (Fig. 4a) is also found realistic when compared with observations (i.e. the Fig. 2b of Jochumsen et al. (2012)).

However, flaws remain regarding the temperature of the deepest waters which are bearly below 1°C when observations clearly show temperatures below 0°C (e.g. Jochumsen et al. (2012), Jochumsen et al. (2015), Zhurbas et al. (2016)). A small bias toward greater salinity values (not shown) is also found in the control experiment which shows bottom salinity of 34.91 compared to 34.9 in the observations shown in Mastropole et al. (2017), but the resulting stratification in density shows patterns that are consistent with observations (Fig. 4b). Although the present setup is designed to investigate model sensitivity in twin

experiments and not for comparison with observations ends, the control run appears to provide a flow of dense waters at the sill that is stable over the 5 year period of integration and reproduces qualitatively the major patterns of the overflow "source waters" seen in the observations. Therefore, despite existing biases, the presence of a well identified dense overflow at the sill confirms the adequacy of the configuration for the sensitivity studies.

If similarities with observations are found at the sill, the evolution of the DSO plume in the Irminger basin is shown to be

unrealistic in the present setup of the control simulation, and presents the same flaws as in the global run. This is demonstrated by the analysis of the temperature and potential density profiles at four different cross-sections along the path of the DSO in the Control simulation (the plots on the left hand side of Fig. 5 and 6). The evolution of the DSO plume as it flows southward along the East Greenland shelf break is represented by a well marked boundary current carrying waters of greater density than the ambient waters. If one identifies the core of the DSO plume by the $27.85$ isopycnal, it is clear that the plume is sinking

to greater depth as it moves southward. This evolution is only qualitatively consistent with the observations at these section





**Figure 2.** *Surface an bottom current speed (year 76 mean) respectively in (a) and (c) the global ORCA12 simulation and (b) and (d) the regional DSO12.L46 regional simulation. One vector every four points is shown. Vectors at the bottom circulation are scaled by a factor of 7 for visibility reasons. Vectors/Colors indicate current direction/speed in $ms^{-1}$.*

Quadfasel (2004). The modelled plume is significantly warmer and exhibits a core temperature of 3.5° (against 2°C or below in the observations). The plume, which is also much wider than observed, exhibits much smaller temperature and salinity gradients separating the plume from the interior ocean, indicating a greater dilution with ambient waters. The plume is barely distinguishable from the ambient fluid below $2000m$ when it is still well marked at that depth in the observations.

5     The bottom temperature shown in Fig. 7a illustrates this excessive dilution of the overflow waters. Indeed, the cold water tongue seen in the bottom of the Denmark strait at a temperature of about 2°C clearly sinks as it extends to the southwest and crosses the 1000 m and the 1500 m isobaths. But as it sinks, it is rapidly diluted and looses its "cold water" character, and is



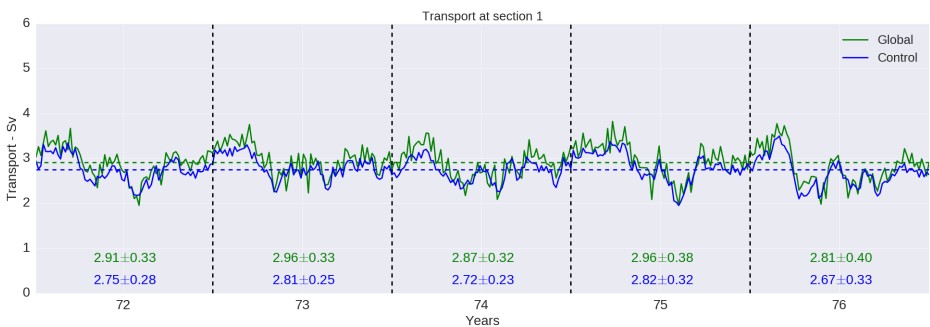

**Figure 3.** *Control Simulation. Time evolution at the sill section (Section 1 in Fig. 1) of the volume transport of waters of potential density greater than $27.80\ kgm^{-3}$. The transport annual mean and std (in Sv) is indicated for every individual year of the Control simulation. The transport for the Global model that provides the open boundary conditions is also shown.*

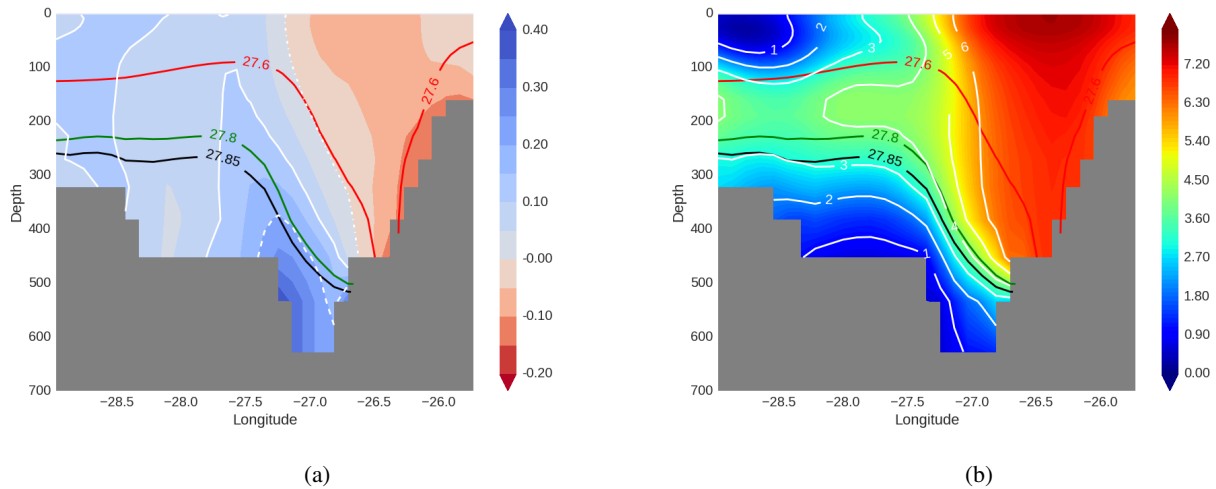

(a)                                                                 (b)

**Figure 4.** *Mean flow characteristics (annual mean of year 76) in the global simulation at the sill. a) The velocity normal to the section in colors (southward velocity being negative). White lines indicate the $0m.s^{-1}$ contour (dotted line), the $-0.1m.s^{-1}$ (full line) and the $-0.2m.s^{-1}$ contour (dashed line). b) Temperature (°C) in colours and white contours. Potential density values ($\sigma_0$) are shown by the contour lines coloured in red (27.6), green (27.8) and black (27.85).*

not distinguishable from the background waters beyond 64.5°N. Such plot of the bottom temperature summarizes rather well what we also learned in the analysis of the cross sections (e.g. Fig. 5 and 6). The same plot for the salinity (not shown) shows waters fresher than surrounding waters in the overflow path, with a salinity that increases from 34.91 at the sill to around 34.96 at 1000 m and reaches the background value ($\sim 35.02$) at 1500 m depth. This demonstrates the large dilution by entrainment

5    with the waters of the Irminger current in the final solution.







**Figure 5.** *Distribution with depth of the annual mean temperature (5th year of model run) at Section 16 (Dohrn Bank array) for a) simulation DSO12.L46, and b) simulation DSO60.L150 (Child grid), and at Section 20 (spill jet section) for c) simulation DSO12.L46, and d) simulation DSO60.L150 (Child grid). Temperature (°C) in colours with white contours. Potential density values ($\sigma_0$) are shown by the contour lines coloured in red (27.6), green (27.8) and black (27.85) The white-rimmed dot on figure (b) correspond to the location of the profiles displayed in Fig. 14 and the red-rimmed and blue-rimmed dots on figure (d) correspond to the locations of the profiles displayed in Fig. 12.*

## 3   Results from sensitivity experiments

We performed a large set of simulations (over 50) with different settings of the DSO model configuration in order to better understand the impact of the different parameters on the final representation of the DSO at a resolution of $1/12°$, including a few with local grid refinement. The detailed list of theses experiments is provided in Appendix A. Following what is used for the Control DSO12.L46 experiment, the simulations are referred as DSOxx.Lyy where $xx$ informs on the horizontal grid



**Figure 6.** *As in Fig. 5 at Section 24 (TTO array) for a) simulation DSO12.L46 and b) simulation DSO60.L150 (Mother grid), and at Section 27 (Angmagssalik array) for c) simulation DSO12.L46 and d) simulation DSO60.L150 (Mother grid).*

resolution (e.g. 36 for 1/36°), and $yy$ on the number of vertical levels (e.g. 150 for 150 levels). After testing different physical parameterizations, numerical schemes and grid resolutions, we concluded that the only parameters affecting the overall representation of the overflow in a significant way are the horizontal and vertical resolutions. No significant impact was found on the representation of the DSO for all the other parameters tested, the flaws described in the previous section resisting the changes.

5    Therefore, we only present the results obtained when the resolution (vertical or horizontal, or both) is changed. The other sets of sensitivity tests are very briefly discussed in Appendix A.

From the large set of diagnostics performed to assess the impact of model changes on the DSO, it was found that the analysis of the bottom temperature in the Irminger Basin is quite a pertinent way to provide a first assessment of the changes



in the properties of the overflow. This diagnostic is consequently used to first compare the different sensitivity simulations, and additional diagnostics are used later for more quantitative assessments of the DSO representation.

## 3.1 Sensitivity to vertical resolution at $1/12°$

(a)

(b)

(c)

(d)

**Figure 7.** *Annual mean bottom temperature (5th year of simulation) in °C at $1/12°$ horizontal resolution in simulations differing the number of vertical levels. a) 46 levels (DSO12.L46) b) 75 levels (DSO12.L75), c) 150 levels (DSO12.L150) d) 300 levels (DSO12.L300). Isobaths 500m, 1000m, 1500m and 2000m are contoured in black.*

The first set of tests that we present is the sensitivity of the DSO representation to the vertical resolution at $1/12°$ horizontal grid resolution. The DSO12.L46 control run (46 levels) is compared with simulations with 75, 150 and 300 vertical levels, all other parameters being identical. The mean bottom temperature of the 5th year of these 4 simulations (Fig. 7) reveals that the





increase in vertical resolution at $1/12°$ works to the detriment of the representation of the overflow. In the 75 levels case, the descent of the DSO plume stops at the 1500 m isobath blocked by a westward flow of warm Irminger waters that invades the 1500 m to 2000 m depth range. This yields a general warming of the bottom waters in the Irminger Basin and along the whole East Greenland shelf break. The overflow representation improves slightly in the 150 levels case as the DSO plume still reaches

the 2000 m isobath, feeding the deep basin, but with less efficiency than in the 46 levels case. Finally, the representation of the DSO is even more degraded in the 300 level case, this resolution exhibiting the greatest dilution of the DSO waters among all resolutions, which was not expected since it should allow for the best resolution of the bottom Ekman layer. This deterioration of the overflow properties was verified in all the other diagnostics (hydrographic sections, T,S diagrams, etc.).

To understand that behavior, one recalls how the cascading of dense water is treated in the z-coordinate NEMO framework. In

case of static instability (i.e. when the fluid at a given level has a greater potential density than the fluid at the next level below), the vertical mixing coefficient, usually calculated with the TKE closure scheme, is assigned a very large value (usually 10 $m^2 s^{-1}$). This instantaneously (i.e. over one time step) mixes the properties (temperature, salinity, and optionally momentum) of the two cells, re-establishing the static stability of the stratification. This parameterisation, referred to as EVD (Enhanced Vertical Diffusion already described in Section 2.1), is at work to simulate the sinking (convection) and the cascading (overflow)

of dense waters. Note that when the EVD was not used in our experiments, we noticed that the TKE mixing scheme often produced values of the diffusion coefficient larger than 1 $m^2 s^{-1}$ and in very particular cases exceeding 10 $m^2 s^{-1}$.

The vertical diffusivity along the path of the overflow is shown in Fig. 8 (the definition and method of calculation of the overflow path are given in Appendix B). Compared to the 46 level case, the 300 level case (Fig. 8b) exhibits greater values of the diffusion coefficient near and above the bottom along the path of the overflow. This enhanced mixing affects the overflow

plume, which 200 km after the sill does not contains waters denser than 27.85 $kg.m^{-3}$, while such waters are still found 300 km down the sill in the 46 level case. An explanation to this is searched for following the paradigm of Winton et al. (1998) which states that the horizontal and vertical resolutions should not be chosen independently: the slope of the grid ($\Delta z/\Delta x$) has to equal the slope of the topography ($\alpha$) to produce a proper descent of the dense fluid (see their Fig. 7). If this is not the case, the vein of dense fluid thicken by mixing with the ambient fluid at a rate proportional to the ratio of the slopes $\alpha(\Delta z/\Delta x)$.

To show the effect of this concept, we simulate the descent of a continuous source of cold water down a shelf break in an idealized configuration of NEMO (with no rotation, comparable to that of Winton et al. (1998)). The configuration (Fig. 9) is as follows. A 20km wide shelf of depth 500m is located on the left side of the 2D domain. It is adjacent to a shelf break wide of 250km reaching the depth of 3000m, and then the bottom is flat. Initial conditions are as follows. A blob of cold water is placed on the bottom of the shelf. Its temperature is 10°C when the temperature of the ambient fluid in the rest of

the domain is 15°C. The salinity is constant and equal to 35 in the whole domain. During the simulation, the temperature is restored on the shelf to its initial value to maintain the source of cold water. A relaxation to the ambient temperature is applied over the whole water column in the last 50km of the right side of the domain in order to evacuate the cold water. The horizontal grid resolution is 5km (comparable to the 1/12° resolution of our regional DSO configuration). Two simulations with different vertical resolutions are run. The first one uses 60 levels of equal thickness (50 m) such that the local grid slope always equals

the slope of the bathymetry (Fig. 9a, $\Delta z = 50m$). In the second simulation, the vertical resolution is increase by a factor of 5







(a)

(b)

**Figure 8.** *Annual mean (5th year of simulation) of the vertical diffusivity coefficient along the path of the vein calculated for a) simulation DSO12.L46 and b) simulation DSO12.L300. Potential density values ($\sigma_0$) are shown by the contour lines coloured in red (27.6), green (27.8) and black (27.85).*

(300 vertical levels, Fig. 9b, $\Delta z = 10m$). In the absence of rotation, the pressure force pushes the blob over the shelf break and the EVD mixing scheme propagates the cold water down to the bottom as the blob moves toward deeper waters, generating an overflow plume. After about 5 days, the front of the plume has reached the end of the shelf break and entered the damping zone at the right side of the domain, reaching a quasi-stationary regime. In that regime, the overflow simulated in the 300 vertical levels run (i.e. with a local grid slope smaller than the topographic slope) presents warmer bottom waters (Fig. 9a) than in the 60 levels run, validating to a certain extent the rationale exposed in Winton et al. (1998) and in agreement with the results obtained with the realistic DSO12 configuration. Note that the vertical shear is more confined in the high-resolution case, which



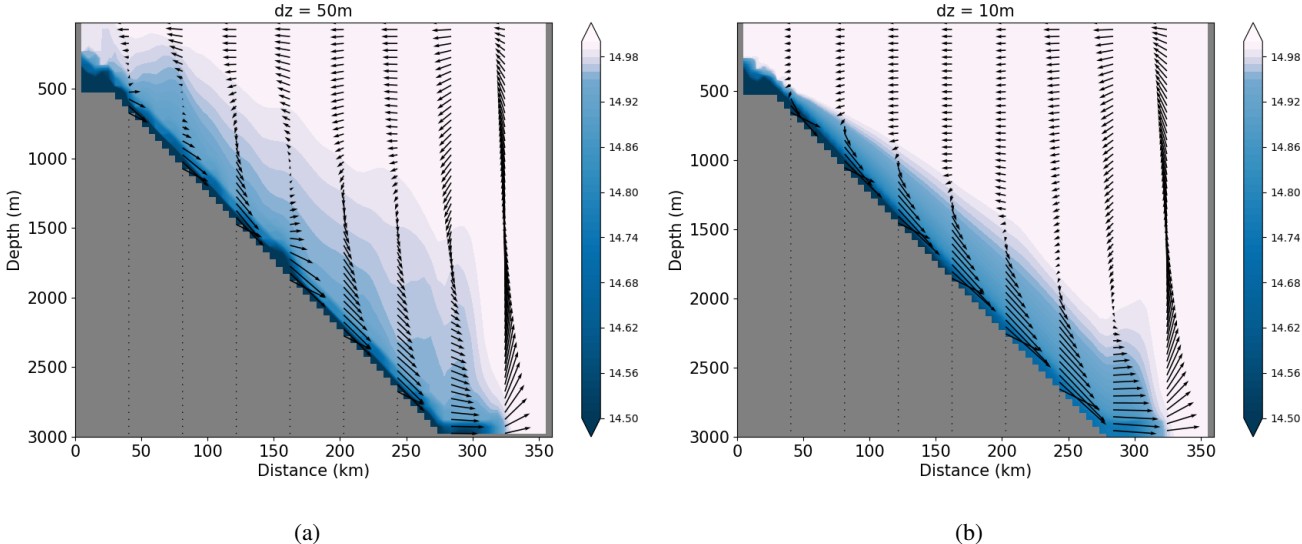

**Figure 9.** *Idealized experiment simulating the rationale exposed in Fig. 7 of Winton et al. (1998). Cold water (10°C) descends a shelf break in a configuration with ambient water at 15°C after 9 days of simulation. Two vertical resolutions are used: a) $\Delta z = 50m$ b) $\Delta z = 10m$. Temperature (°C) in colors. Vectors represent the velocity, the vertical velocity being re-scaled according by the grid aspect ratio of case (a).*

prevents the upward extent of the TKE induced mixing of the upper part of the overflow that is seen in the low resolution case. Note that when using a realistic bottom topography, the topographic slope will present large local variations and that it will be almost impossible to match the two slopes over the whole domain in a z-coordinate context. Therefore, increasing the number of vertical levels will not systematically degrade the overflow representation everywhere.

A set of simulations tested the effect of the different closure (i.e. vertical mixing) schemes available in NEMO (TKE with and without EVD, $k - \epsilon$ with and without EVD, constant diffusivity+EVD, Madec et al. (2016)). It was found that the choice of the vertical mixing scheme has a very small (insignificant) impact on the final representation of the overflow at $1/12°$.

### 3.2 Sensitivity to a local increase in horizontal resolution

#### 3.2.1 The $1/36°$ case

The second set of tests that we present is the sensitivity of the DSO representation to the vertical resolution using a local horizontal refinement of $1/36°$ in the overflow region (see Fig. 1). The same range of vertical levels as for the $1/12°$ resolution case is investigated: 46 levels (DSO36.L46), 75 levels (DSO36.L75), 150 levels (DSO36.L150) and 300 levels (DSO36.L300). The annual mean bottom temperature of the 5th year of simulation is shown in Fig. 10. Compared to the $1/12°$ cases the $1/36°$ cases present, at equivalent number of vertical levels, significantly colder bottom temperatures in the Irminger basin and

along the East Greenland shelf break. This amelioration is rather small at 46 levels (Fig. 10a, the cooling is $\sim 0.4°C$ ), but is



more significant for the other vertical resolutions. The greatest improvement is observed when the number of vertical levels is increased to 150 levels (Fig. 10c). In this case the signature of the DSO becomes evident. The bottom temperature of the overflow plume in its first 100 km cooled from a value of $\sim 3.6°C$ at $1/12°$ (Fig. 7a) to a value of $\sim 2.7°C$ (a remarkable cooling of $\sim 0.9°C$) while the temperature at the sill did not change.

(a)

(b)

(c)

(d)

**Figure 10.** *Annual mean of the bottom temperature of year 5 of simulations using grid refinement $(1/36°)$ a) DSO36.L46 b) DSO36.L75 c) DSO36.L150 d) DSO36.L300. Isobaths 500m, 1000m, 1500m and 2000m are contoured in black.*

5    The situation changes when increasing to 300 levels (Fig. 10d). The tendency for improvement noticed when increasing from 46 to 150 levels is reversing and the representation of the overflow is slightly degraded. This result is coherent with the explanation given for the $1/12°$ case. Once reached a vertical resolution that is adequate for a specific horizontal resolution for a given slope, increasing the vertical resolution will deteriorate the DSO representation by introducing excessive vertical





mixing. A relevant remark here is that over-resolving the slope in the vertical is in detriment of the overflow to represent, which is consistent with the conclusions of Winton et al. (1998). In other words, it exists an optimal number of vertical levels to be used for a given horizontal resolution for a given slope. Given the large variety of slopes present in the oceanic topography (and encountered by an overflow during its descent), modelling topographic constrained flows with z-coordinates appears as a

quite difficult task.

### 3.2.2  The $1/60°$ case

Continuing with our rationale, we evaluate the representation of the DSO at $1/60°$ (using a local refinement in the area shown in Fig. 1) with 46, 75, 150 and 300 vertical levels. At this resolution, the 46 levels and 75 levels cases shows solution very similar to that presented at the resolution of $1/36°$ (no significant additional improvement, no figure shown). A significant

change is again observed for 150 levels (Fig. 11a). The signature of the overflow waters at the bottom is even stronger in this case, the cooling of the overflow plume being $\sim 1.1°C$ when compared to the $1/12°$ solution (Fig. 7a), and $\sim 0.2°C$ compared to the $1/36°$ case.

The solution of the 300 levels case at $1/60°$ (Fig. 11b) represents an improvement compared to the $1/36°$ case with the same vertical resolution. However, compared to the $1/60°$ and 150 levels solution (Fig. 11a) it shows a slightly greater dilution

of the overflow and warmer temperatures at the bottom. Also the propagation of the dense water away from the refinement area is clearly better with 150 levels. This should be taken into consideration when choosing the refinement region if used in global implementations. Improvements brought to the representation of the overflow by the resolution increase to $1/60°$ and 150 levels can also be seen in Fig. 5 and 6 and is quantitatively assessed in the following section (Section 4). At every section, the DSO60.L150 overflow (right hand side plots) is clearly identified by a vein of cold waters well confined along the

slope with temperatures below 3°C and always at least 0.5°C colder than in the reference simulation (DSO12.L46, left hand side plots). Temperature gradients between the core of the overflow and the interior ocean are also significantly increased, and the isopycnal 27.85 marks very well the limit of the vein of fluid. If a warm bias still exists compared to the observations of Quadfasel (2004) (bias for a part due to the unrealistic properties of the interior entrained waters), the agreement of the overflow pattern with the observations is nevertheless greatly improved.

## 4  Eddy-resolving solution ($1/60°$ grid and 150 levels)

The worsening of the DSO representation with increasing vertical resolution until a certain extent is observed with the three horizontal resolutions used in this study ($1/12°$, $1/36°$ and $1/60°$). The analysis of the high vertical diffusivity values due to the EVD demonstrated the dominant impact of this parameterization on the overflow at the resolution of $1/12°$ and emphasized the need for coherent vertical and horizontal grids. However, the improvements observed in both the DSO36.L150 and

DSO60.L150 cases suggest that this impact is reduced and other drivers take control the evolution of the overflow plume when higher horizontal grid resolutions are used. To reach a better understanding of the reasons for improvement, we perform in this section an analysis of the overflow structure in the $1/60°$ and 150 level simulation (DSO60.L150).

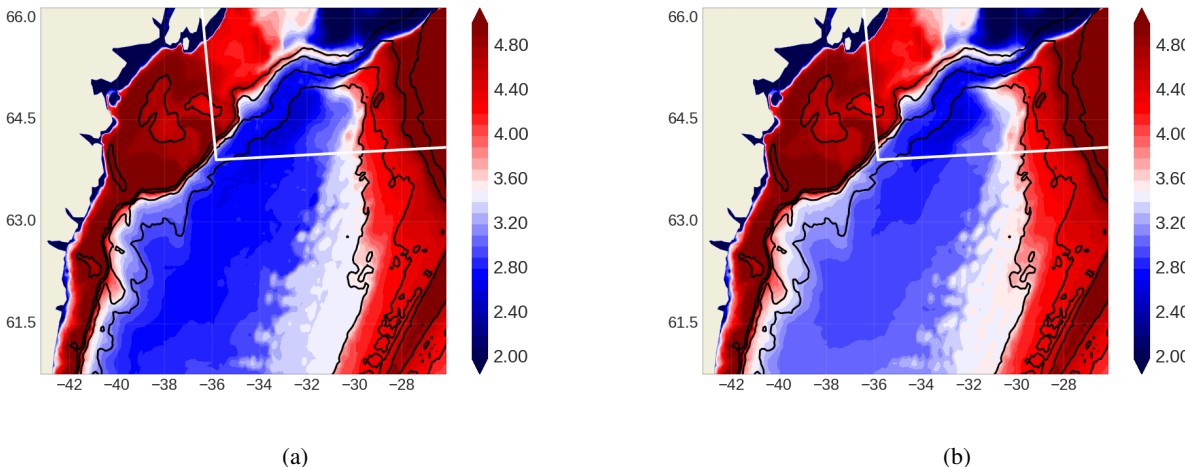

**Figure 11.** *Annual mean of the bottom temperature in the 5th year of simulations using a grid refinement of $1/60°$ for a) 150 vertical levels (DSO60.L150) and b) 300 vertical levels (DSO60.L300). Isobaths 500m, 1000m, 1500m and 2000m are contoured in black. The white box indicates the area of refinement.*

Fig. 12 shows vertical profiles of the mean along slope velocity at two different locations on the shelf break at section 20 from four different simulations which use a large number of vertical levels (150 or 300). All profiles, except that of the 1/12° with 150 level case, show a bottom intensified boundary current confined in the first $200m$ above the bottom, and the presence of a sheared bottom Ekman layer better resolved with 300 levels but still well marked with 150 levels. This indicates the presence

of a well defined overflow plume, as shown for the DSO60.L150 in Fig. 5d. This bottom signature has already been described in observations (Paka et al. (2013)). In the other $1/60°$ cases with lower vertical resolution (DSO60.L46 and DSO60.L75, not shown) the Ekman driven vertical shear cannot be resolved and the whole dynamics of the current is dominated by the EVD mixing.

The absence of this bottom-confined intensified current in the DSO12.L150 simulation can be related to a similar cause,

although the vertical resolution is sufficient to partially resolve the Ekman bottom layer. The analysis of the vertical mixing coefficient (Fig. 13b,13d) shows a very intense mixing in a rather thick layer all along the slope (between $500m$ and $2200m$) and, according to the previous rationale, the reason is the convective adjustment (EVD) governing the near bottom physics. This enhanced mixing seriously limits the development of a sheared flow in the bottom layer.

In the case of DSO60.L150 (Fig. 13a,13c) the EVD driven mixing remains confined to a very thin bottom layer below the

15 27.85 isopycnal, and very little mixing occurs in the core of the overflow plume. Intermittent static instabilities occur between the 27.85 and the 27.8 isopycnals, the associated mixing being small since the temperature and salinity gradients are quite small there. This behavior is consistent with the physical processes present in the DSO, the simultaneous action of the shear governing the entrainment in the overflow plume and the density gradient driving the overflow to the bottom. In this way, the use of coherent horizontal and vertical resolutions plays a key role since it allows the convective adjustment to occur in

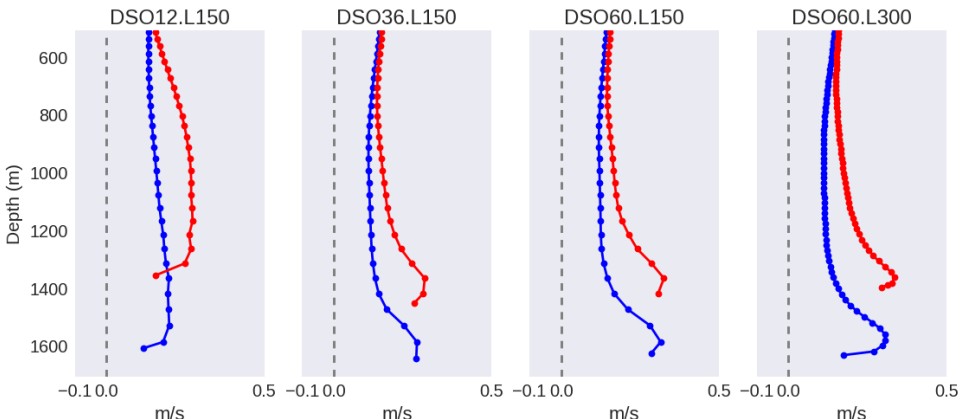

**Figure 12.** *Vertical profiles (annual mean) of the quasi along-slope velocity (ie. velocity normal to the section) at the two locations of section 20 indicated in figure 5d for simulations (a) DSO12.L150, (b)DSO36.L150, (c) DSO60.L150, and (d) DSO60.L300.*

a limited portion of the plume without interfering with the other important processes driving the physics in the vein of dense fluid. We identify then three conditions for a proper representation of the DSO: coherent vertical and horizontal grids to avoid excessive convective adjustment (due to EVD or any other scheme); proper vertical resolution to resolve the shear induced by the Ekman layer dynamics; and enough horizontal resolution to resolve the boluses of the DSO (described afterwards). This

agrees with what is stated in the idealized study of Laanaia et al. (2010), it is not the increase of vertical viscosity that enables the down-slope movement, but the resolution of the bottom Ekman layer dynamics.

Continuing with the description of the bottom flow, we show in Fig. 14 the vertical profiles of physical properties at a specific point at section 16, chosen close to where Paka et al. (2013) performed microstructure measurements (65.20°N 30.41°W) in order to allow for a direct comparison. Our DSO60.L150 simulation reproduces with a high degree of realism the main features

of the observed plume (as shown in Fig. 4 of Paka et al. (2013)). As in the observations, the plume is nearly $200m$ thick and is located between $1200m$ and $1400m$ depth. Compared to the water above it, the modelled plume is characterized by a freshening of $\sim 0.15$ ($\sim 0.10$ in the observations) and a cooling of $3.0°C$ ($\sim 3.5°C$ in the observations). The cross-slope and along-slope velocities show an acceleration of the flow in the plume of $\sim 0.2ms^{-1}$ and $\sim 0.6ms^{-1}$ respectively in observations, the corresponding values in the model being $0.3ms^{-1}$ and $0.7ms^{-1}$. Since this $200m$ thick plume is represented by 5/6 points in

the vertical, the use of 150L might be a lower bound for the number of vertical levels to use in order to properly represent the DSO.

The boluses of cold waters mentioned in different observational and modelling papers (see for example Girton and Standford (2003), Jochumsen et al. (2015), Magaldi et al. (2015), Koszalka et al. (2017)) are also reproduced by the model. To illustrate this we show in Fig. 15 hourly outputs of the bottom temperature in a sequence that lasts only $\sim 40$ hours. First, in figure

15a the DSO appears as a cold water ($\sim 1°C$) plume that has already started its descent and is confined between the $500m$ and $1000m$ isobaths. Boluses of cold water ($\sim 2°C$) are also seen a few tens of $km$ downstream in the depth range $1500m$ to





(a)

(b)

(c)

(d)

**Figure 13.** *Vertical diffusivity coefficient (hourly mean) at section 20 in simulations a,c) DSO60.L150, and b,d) DSO12.L150 in winter time. In black: contours of isopycnals 27.6, 27.80 and 27.85.*

$2000m$ and in the deep Irminger Basin. Fifteen hours later (Fig. 15b), the DSO plume has sunk to $1500m$, seems to be adjusted to geostrophy and flows along isobaths. Another plume of cold water is moving through the sill. In the following 24 hours, the first plume moves along the shelf break (Fig. 15c) and breaks into a bolus which brings cold waters to the depth of $2000m$ (Fig. 15d). A significant entrainment of surrounding waters occurred during the breaking as the water in the bolus has gained

5 about $0.5°C$. The bolus will continue its way to the Angmassalik array, i.e. section 29 in Fig. 1, and will contribute to cool the deep Irminger Basin. The second plume has crossed the sill and reached the $1000m$ isobath. It will later generate another bolus following the same process. The formation of the bolus happened in only 40 hours, showing the high frequency variability of the overflow and illustrating the difficulty of diagnosing its time-mean properties.





We attempted a quantitative comparison with the relatively long-term observations made at the mooring arrays reported in the study of Voet and Quadfasel (2010). These arrays are the Sill Array (section 1 in Fig. 1), the Dorhn Bank Array (section 16 in Fig. 1), the TTO Array (section 24 in Fig. 1) and the Angmagssalik Array (section 29 in Fig. 1). Following Voet and Quadfasel (2010) we reported in (Fig. 16) the minimum time-mean bottom temperature at these four sections for certain

simulations. This figure somehow summarizes our main findings. In the $1/12°$ simulations, the temperature at the mooring arrays (i.e. the dilution of the overflow) increases with increasing vertical resolution, DSO12.L46 showing a lesser dilution for the first $200km$ of the overflow path than DSO12.L150. The best performing simulation at $1/36°$ is that with 150 levels (DSO36.L150). It shows a cooling of the bottom waters after the sill of about $0.5°C$ when compared to DSO12.L46. At $1/60°$ resolution, the best performing simulation is also that with 150 levels (DSO60.L150). When compared to the best $1/12°$

simulation (DSO12.L46) it shows an even greater cooling of the bottom waters after the sill ($0.7°C$). Increasing the vertical resolution to 300 levels produces a slightly greater dilution of the plume that could be considered as insignificant, but the computational cost is doubled. When comparing this set of best performing simulations with observations, it appears that the model always produces a much greater dilution of the physical properties of the overflow in the first $200km$ of its path. Improved initial conditions (i.e. correcting for the warm bias of $0.3C$ at the sill and for the warm and salty bias of the entrained

waters of the Irminger Current) will certainly reduce this difference but to a point which is difficult to estimate. Either way, the 1.5°C difference shown in Fig. 16 is a quite wide gap to fill.

Finally, we would like to point out that the increase in resolution also improves the representation of topography. For example, the thin v-shaped channel over the sill (Fig. 4) is better represented as resolution increases. This leads to a more separated cold and fresh DSO current from the warm and relatively salty Irminger current, specially during the descent.

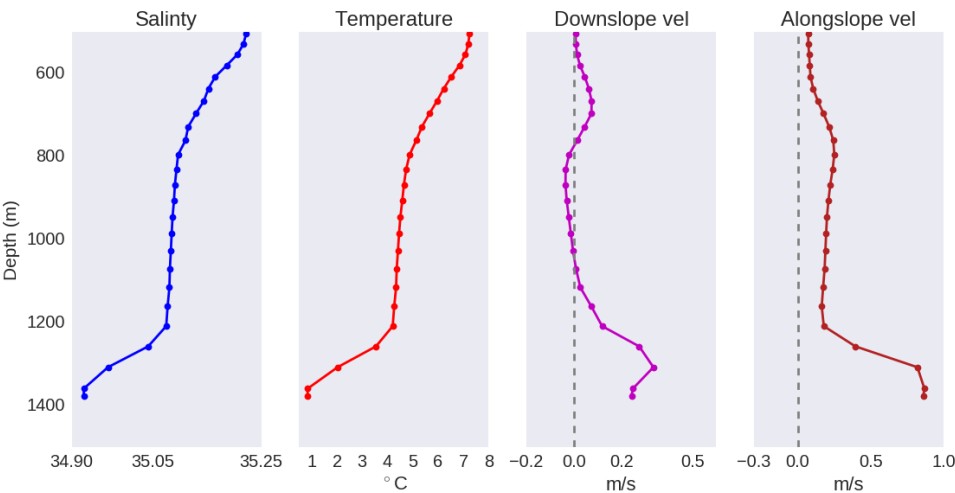

**Figure 14.** *Vertical profiles of the physical properties of the overflow plume in simulation DSO60.L150 (hour 10 on 01/February of year 5) at section 16 for the location indicated with a white-rimmed dot in Fig. 5b. a) Temperature, b) Salinity, c) cross-slope velocity, and d) Along-slope velocity.*



**Figure 15.** *Hourly snapshots of bottom temperature in simulation DSO60.L150 for 4 January a) at 14h, b) 15h after, c) 26h after, d) 39h after. The 500m, 1000m, 1500m and 2000m isobaths are contoured.*

## 5 Summary and Conclusions

We evaluated the sensitivity of the representation of the Denmark Strait overflow in a regional z-coordinate configuration of NEMO to eddy-permitting to various eddy-resolving horizontal grid resolutions (1/12°, 1/36° and 1/60°), the number of vertical



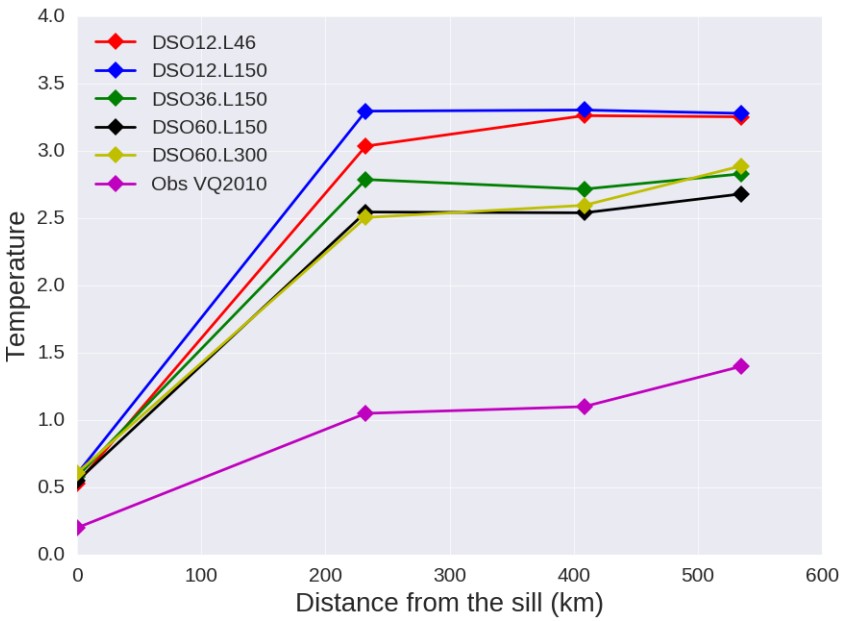

**Figure 16.** *Time mean minimum bottom temperature for five sensitivity simulations at the location of the observational arrays. The results of Voet and Quadfasel (2010) are reported as Obs VQ2010.*

levels (46, 75, 150, 300), and to numerical and physical parameters. The first unexpected result was that the representation of the overflow showed very little sensitivity to any parameter except the horizontal and vertical resolutions. Contrary to expectations, in the given numerical set-up, the increase of the vertical resolution did not bring any improvement when an eddy-permitting horizontal grid resolution of $1/12°$ (i.e. $\sim 5km$) is used. We found a greater dilution of the overflow as the number of vertical

level was increased, the vein of current becoming warmer, saltier and shallower, the worse solution being the one with 300 vertical levels. Thanks to a point-wise definition of the center of the vein of current we were able to diagnose the vertical diffusivity along the path of the overflow. Our results show that, as expected in a z-coordinate hydrostatic model like NEMO, the sinking of the dense overflow waters is driven by the enhanced vertical diffusion scheme (EVD) that parametrizes the vertical mixing in case of a static instability in the water column. But our analysis showed that the smaller the local grid

slope when compared to the topographic slope, the more diluted the vein. Since for a fixed horizontal resolution the grid-slope reduces as the number of vertical levels increases, the overflow is more diluted when a large number of levels is used. To limit this effect, an increase of the vertical resolution must be associated to an increase in the horizontal grid resolution.

We then tested the effect of increasing the number of vertical levels (46, 75, 150 and 300) as it was done at $1/12°$ but with an eddy-resolving horizontal grid resolution of $1/36°$ ($\sim 1.5km$). While only slight improvements were found for the

46 and 75 levels cases, the 150 levels case presented a drastic improvement. At such horizontal and vertical resolution the EVD convective adjustment associated with the step-like representation of the topography remained limited to a relatively thin





bottom layer representing a minor portion of the vein. The increase to 300 levels caused a slight deterioration of the DSO representation, generating an increase of the EVD convective adjustment, and being therefore an excessive number of vertical levels for $1/36°$.

Finally, we performed the same series of sensitivity tests with a horizontal grid resolution of $1/60°$ ($\sim 1km$). With 46 and
75 levels, no appreciable differences where found with the correspondent cases at $1/36°$. The 150 levels solution showed an improvement even greater than at $1/36°$, being the most performing of all simulations presented in this work. The increase to 300 levels at $1/60°$ was again in detriment of the DSO representation for the same reasons a explained above. As for the $1/36°$ case, the EVD in the 1/60° and 150 levels case remained limited to a thin bottom layer representing a minor portion of the vein which limited the dilution of its properties. The major additional drivers of the sinking of the overflow at eddy-
resolving resolutions are the generation of mesoscale boluses of overflow waters and the appearance of a resolved vertical shear that results from the resolution of the dynamics of the Ekman boundary layer. Combined to the isoneutral diffusion in the equation for tracers this allows a proper calculation of the entrainment by the TKE scheme and a significant improvement of the properties of the overflow waters.

One interesting conclusion of this work is that for most of the cases tested the EVD convective adjustment was the main pa-
rameter controlling the dynamics of the overflow and becoming the dominant player of the vertical mixing scheme. Indeed, the importance of many other numerical parameters were tested (momentum advection scheme, lateral friction, BBL parameteri- zation, etc.) but none had a significant impact on the overflow representation. Moreover, our results show that the problematic of modelling the overflows is not only about resolving the driving ocean processes, but also about how the grid distribution copes with the phenomena to represent. The rationale that we proposed here is that the horizontal and vertical grid resolutions
necessary to achieve a proper representation of the dynamical processes driving the overflows must be adjusted to be coherent with the slopes along which the overflow descends to limit the vertical extent of the vertical mixing that ends up deteriorating the final solution.

This conclusion draws attention to a limitation for future global simulations in z-coordinate since all flows that are topo- graphically constrained do not flow along the same topographic slope. The model setting for which we obtained our best
representation of the Denmark Strait overflow might not be suitable for an overflow in another location.

All in all, the best results were achieved with the local implementation in the overflow region of the two-way refinement software AGRIF at $1/60°$ with 150 vertical levels. With this drastic increase in horizontal and vertical resolution, among the highest to our knowledge in this type of study, we were able to at least partly resolve the bottom boundary layer dynamics and to simulate an overflow with properties comparable with those seen in the observations. However, significant discrepancies
remained between the model and the observations, being possibly attributed to biases in the initial conditions, the overflow waters being too warm at the sill and the ambient waters entrained in the overflow being too warm and salty at the beginning of the simulations.

For a given vertical number of levels the cost of the implementation of AGRIF in this regional 1/60° configuration case was around 70 times the original cost at 1/12° resolution. Even if this implementation was effective and considering that
smaller proportional costs are expected in configurations of larger domains, this appears as a computationally costly option.



We therefore concluded that a more suitable solution should be searched for. In on going following studies we investigate the representation of the Denmark Strait overflow in a local implementation in NEMO of a terrain following s-coordinate.

*Code and data availability.* The code of the model corresponds to revision 6355 of NEMO v3.6 STABLE (see Madec et al. (2016) for more information), under the CeCILL licence. It can be downloaded from https://zenodo.org/record/3568221. The namelists and the post-

processing scripts can also be downloaded from the same link. The data used to initialize and perform the simulations can be downloaded from https://zenodo.org/record/3568244 (1/12° and 1/36° horizontal resolution simulations) and https://zenodo.org/record/3568283 (1/60° horizontal resolution simulations). Model outputs and diagnostics are available upon request.

## Appendix A: Summary of Experiments

We list here the experiments that we performed before arriving to the conclusions described on this paper. For each experiment

we present the main findings in a very succinct way.

**Experiment 1**: Impact of BBL with vertical resolution, Full and partial steps at $1/12°$. Set of 12 simulations combining the possibilities of 46L, 75L and 300L with and without BBL, with partial steps or full steps. Additional tests with 150L and 990L in partial steps without BBL were performed. The variations with depths of the vertical levels is shown in Figure A1. The 46 levels vertical grid uses 29 levels in the first 2000 m and has a cell thickness of 210 m at that depth. The 75 level vertical grid

uses 54 levels in the first 2000 m and has a cell thickness of 160 m at that depth. The 150 levels vertical grid uses 104 levels in the first 2000 m and has a cell thickness of 70 m from that depth. The 300 levels vertical grid uses 160 levels in the first 2000 m and has a cell thickness of 22m at that depth. The main findings are:

- Partial step is more performant than Full step no matter the vertical resolution or the use of BBL.

- More diluted waters when used BBL. Attributed to the grid direction of the sinking of waters (rather diagonal)

- More diluted waters with increasing vertical resolution

**Experiment 2**: Impact of vertical mixing scheme at $1/12°$ 300L. Five runs: TKE with and without EVD, background diffusivity only with EVD, $k - \epsilon$ with and without EVD. All solutions were extremely similar. After studying the whole set of diagnostics, we then concluded that the main driver of the descent of the DSO at $1/12°$ was the presence of high vertical diffusivity values due to density inversions.

**Experiment 3**: Impact of vertical resolution (46L, 75L and 300L) at $1/60°$ with UBS and EEN advection scheme. The use of the UBS scheme did not bring any significant different regarding the solution with the EEN scheme.

**Experiment 4**: Use of EVD on tracers only and on tracers and momentum using 46L, 75L and 300L with an horizontal resolution of $1/12°$ and $1/60°$. No significant changes observed.

**Experiment 5**: Free-slip and No-slip lateral boundary conditions using 46L, 75L and 300L with an horizontal resolution

of $1/12°$ and $1/60°$. No-slip lateral boundary condition shown to improve to some extent the feeding of cold waters to the





Irminger basin as expected (Hervieux (2007)). However, caution must be taken since it has already been shown that this lateral condition can deteriorate the overall global circulation (Penduff et al. (2007)). Only very local treatment approaches must be considered.

**Experiment 6**: Use of Non-Penetrative Convective adjustment instead of EVD at $1/12°$ with 46L and 75L. Almost no differences with EVD, this is believed to be due to the convective adjustment treatment included in the TKE scheme (as in Experiment 2 for 300L).

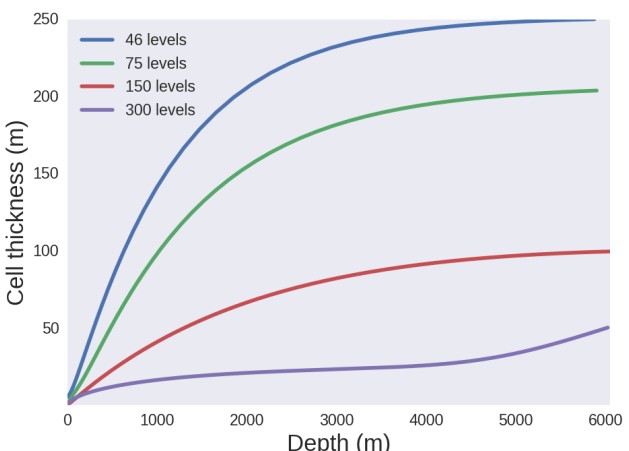

**Figure A1.** *Grid cell thickness as a function of depth for the 4 vertical grids used in the study.*

## Appendix B: Path of the DSO - Calculation

Dickson and Brown (1994), used the density criteria $\sigma_\theta \geq 27.8$ to characterize the DSO considering that this value covers the range of water masses that forms the North Atlantic Deep Waters (NADW). In addition, for the Dohrn Bank, TTO and Angmassalik arrays Dickson and Brown (1994), used a southward velocity greater than zero as an additional criteria in order to guarantee that the water mass considered is effectively flowing in the southward direction. This criteria seems reasonable, since the observational arrays included a large part of the Irminger basin in which deep flows might have northward direction, and would therefore be wrongly considered as part of the DSOW. Brearley et al. (2012), used geographical and density criterias specific to each hydrographic section to define the vein of fluid in their hydrographic sections. Girton and Standford (2003), calculated the center and depth of the overflow by calculating the center of mass anomaly along a number of hydrographic sections. For each section, they limited the extent of the overflow to a width where $50\%$ of the mass anomaly is contained. On the modeling side Koszalka et al. (2013), pointed out the problem represented by the use of the density alone to characterize the overflow, by affirming that the temperature and salinity transformation downstream of the Denmark Strait are not yet well





quantified. To tackle this issue they proposed a complementary description of the overflow by using Lagrangian particles in an offline integration. While this method could be useful to answer some questions, its link with observations is not direct.

In this context we understand that a main characteristic that is not being taken into account for a vein of fluid that is characterized by a large transport is its velocity. However, finding a correct threshold for the southward velocity that works

both for observations and model outputs is not an easy task. Of course this value has to be greater than zero. We might go a bit further and think that we probably should avoid including small velocities related to eddy processes in the Irminger Basin. On the work of Fan et al. (2013) observations were made of the mean peak azimuthal speeds for the anticyclones present in the Irminger basin, obtaining a value of $0.1 ms^{-1}$.

From this point of view any discussion considering a lower threshold value for the overflow should start from at least

$v \leq -0.1 ms^{-1}$. We propose here a value of $v \leq -0.2 ms^{-1}$ because we obtained very robust results. However intermediate values between $-0.1 ms^{-1}$ and $-0.2 ms^{-1}$ can be tested. We then propose a definition similar to the one given by Girton and Standford (2003) for horizontal and vertical positions of the DSO, doing so we define our understanding of the vein and its center.

$$X_{DSO} = \frac{\iint vx\,dz\,dx}{\iint v\,dz\,dx}; (\sigma_0 \geq 27.8, v \leq -0.2 ms^{-1}) \tag{B1}$$

$$Z_{DSO} = \frac{\iint vz\,dz\,dx}{\iint v\,dz\,dx}; (\sigma_0 \geq 27.8, v \leq -0.2 ms^{-1}) \tag{B2}$$

Compared to the definition used in Girton and Standford (2003), we use the local depth of the grid point as a weight instead of the local value of the total depth. As said before, the velocity was also added as a parameter to weight the position of each point. The value of $X_{DSO}$ and $Z_{DSO}$ give the horizontal and vertical position of the core of the vein for each section in

particular.

*Author contributions.* PC, BB, TP, JC, J-MM where part of the design of the experiments and diagnostics, as well as their evaluation. JD, JLS, PV, SG and A-MT where part as well of the diagnostic evaluation. PC and BB wrote the document. PC performed the experiments and performed the diagnostics with the support of JC and J-MM. All authors provided scientific input.

*Competing interests.* The authors declare that they have no conflict of interest.

*Acknowledgements.* PC was supported by UGA though an assistantship from Ministère de l'Enseignement supérieur, de la Recherche et de l'Innovation, and is now supported by CNRS through a grant from CMEMS. BB, J-MM, TP, JLS, and A-MT are supported by CNRS. JC is supported by Mercator Ocean International. Research leading to these results benefited from supports provided by the LEFE/GMMC program



of INSU, the PHC Kolmogorov No 38102RF and the project 14.W03.31.0006 of the Russian Ministry of Science and Higher Education (BB, SG and PV). This work also benefited from many interactions with the DRAKKAR International Research Network (IRN) established between CNRS, NOCS, GEOMAR, and IFREMER. Most important, this research was granted access to important HPC resources under allocations A0030100727 and A0050100727 attributed by GENCI to DRAKKAR, simulations being carried out at the CINES supercomputer

5  national facilities.





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
