# Peer review of "Representation of the Denmark Strait Overflow in a z-coordinate eddying configuration of the NEMO (v3.6) ocean model: Resolution and parameter impacts"

_Geoscientific Model Development, 2019_

## Referee Comment (RC1) · Anonymous Referee #1 · 3 Feb 2020

General comments:

This paper investigates the sensitivity of the Denmark Strait Overflow to horizontal model resolution and number of vertical levels in a regional ocean model based on a global ORCA12 configuration using NEMO. The horizontal resolution varies between 1/12, 1/36 and 1/60 degree using a two-way nesting scheme for 1/36 and 1/60. Vertical levels vary between 46, 75, 150 and 300. It has been found that if the slope of the model grid is smaller than of the actual ocean topography the overflow water entrains too much surrounding water and gets more diluted. Increasing the number of

vertical levels is therefore not always a good choice to improve overflows. This is paper presents a timely topic and a comprehensive modelling study. My comments are only of minor nature to improve the flow of the paper and figures.

Specific comments Introduction:

2-7 High salinity shelf water which is a source for Antarctic Bottom Water is an overflow too and could/should be mentioned here. Around Antarctica most models struggle to get the dense water from the shelf into the abyssal ocean without entraining too much surrounding water.

Methods:

Figure 1. As far as I can tell, only section 29, 24, 20, 16 and Denmark Strait have been used. I do not see much value showing all the other sections. I suggest reducing them to the once which are being shown. I am aware that they are meant to show DSOW core. Please see my comment how alternatively the DSOW could be tracked, which would not require individual sections. Figure 2. It is hard to compare those fields. I would suggest showing the mean from the global configuration and anomalies to the regional setup. In this case it becomes clearer where the differences are. Since both models use the same grid calculating anomalies should be easy. All the subsequent figures have a lot of white spaces between the subplots. If there is any chance to move subplot labels into the figures that would allow to reduce the white spaces and improve the visibility/readability of the figures. 8-14. It appears that the DSOW has a seasonal cycle, which is not present in observations in the Denmark Strait (Jochumsen et al. 2012). Although this is not too critical for this study it shows that likely the formation regions of DSOW in the Nordic Seas are not captured correctly (Våge et al. 2013). That could explain why the transport variability is so low. The seasonal signal usually originates from the EGC and Fram Strait. Figure 4. I would swop (a) and (b) so you can avoid starting in line 8-16 with Figure 4b and later going back to Figure 4a. Figure 5-6. Is there the chance to include observational values here (CTD casts) along some of

these sections? That would help to illustrate how the solution should look like. Maybe just adding density contours would/could already help. 9-1 It remains unclear where this statement is based on, as far as I can tell observations along these sections are not shown or provided. I recommend a re-write of section 8-29 until the results section. The main point is not clear to me. Is it that in the control simulation the temperature in the DSOW layer are more diluted than in the other simulation? If so, this should go in the results section and would also help avoid talking about Figure 7 twice.

Results:

Figure 7. Is it necessary to show the "warm" >3.6°C waters? It distracts from the cold DSOW in the Irminger Sea and would allow to get a bit more structure in these plots. Have you tried using anomalies plots here, to make the point clear that with more vertical levels the bottom water gets eroded? Figure 7,10,11 I think it would help to overlay the DSOW path in these simulations. As the authors stated the DSOW is characterised by a temperature minimum, so the path in these simulations could be also defined by the zonal minimum in the regional for each latitude, an alternative way to what the authors use at present. 22-14 I am not convinced that reducing the model bias in the source waters will help. Results in Figure 16 show that even if modelled temperature would agree with observations, temperatures downstream would end up being warmer than the observations.

Technical corrections:

I could not spot any typos but hope a native speaker might help.

---

## Referee Comment (RC2) · Anonymous Referee #2 · 6 Feb 2020

I think it is an interesting manuscript that deserves to be published after some editing.

The story, if summarized, is that one should be 'resolving' the topographic slope in the sense that the aspect ratio dz/dx of mesh cells is higher than the slope, and that vertical mesh resolution has to be sufficient to represent the plume (in this manuscript 150 layers provide several points (5-6) across the overflow plume in vertical direction).

1. My main problem with the manuscript in its current form is that this story is presented as something unexpected and not known. This starts from the abstract and is repeated

several times in the text. However, at least as concern the dz/dx ratio, the limitation on this ratio is well known (and authors themselves mention several papers). The second aspect is also general enough to be surprising, of course, the overflow plume has to be resolved vertically, there is no hope on representing the overflow otherwise. The statements like "Contrary to expectations ..." are strange in this context, it is, in contrast, in agreement with expectations.

The value of the manuscript is not in the fact that it finds something new and unexpected ("It is found that when the local slope of the grid is weaker than the slope of the topography the result is a more diluted vein" - Is not this known?), but in exploring and documenting precise limitations for the particular ocean circulation model, which will be appreciated by the NEMO community and very likely by other ocean modeling groups.

I would recommend that the authors look critically at their statements and adjust the manuscript accordingly (the Abstract, introduction, conclusions in the first turn). I do not think the present form is acceptable.

2. Even in higher resolution runs the bottom topography was kept from 1/12 degree case, and question arises as what will happen if the topography were adjusted according to the resolution. I would appreciate some discussion of the aspect of resolving the topography. For example, what would happen if 1/12 degree simulations were run on a smoother topography? This might add some useful insight.

3. The manuscript is well written, however it tends to overdefine and at too many places phrases could be more concise. Some editing would be good at this level, but it is up to authors.

Some small issues (not all)

page 2

line 7 check citation style

23 'at that resolution' – which one? Can be removed

page 3

line 4 'yield to'???? The entire sentence can be written as:

The first complication arises from the the neglect of vertical acceleration in the hydro-static approximation leading to misrepresentation ... (see 3 above)

line 30 remove , after (2009)

page 4

lines 8 and 12 'Despite' and then again 'despite'

29 'is presented in' – contains

page 5

line 24 citation style

page 6

line 4 citation style

Caption to Fig.2 an –> and; Surface (a,b) and bottom (c, d) current speed (year 75) in the global ORCA12 (a,c) and regional DSO12.L46 simulations. Only every fourth point is shown....

page 11

lines 4,5 Following the convention for DSO12.L46, the simulations ...

page 14

line 9 Is NEMO different from all others?

line 24 your formula does not express the ratio.

line 28 250 km wide

line 29 when?

page 18

line 1 over-resolving the slope vertically worsens the overflow representation

line 2 there exists or there is

line 7 Which rationale is meant?

page 20

line 13 acceleration? or speed-up (units are of velocity)

line 14 5 - 6 points

---

## Referee Comment (RC3) · Anonymous Referee #3 · 14 Feb 2020

The paper is important, numerical experiments are well designed and carefully performed and written. However the manuscript, to my mind has some major shortages:

1. Absence of figures with observations. It is very hard to follow the text, when the authors refer to figures in the other papers. I found only one plot (Fig 16) to be very informative. Is it possible to plot similar figures from observations? I believe that most of the observed data are present in databases such as EN4.

2. Secondly the introduction of the manuscript is not satisfactory written, the style and

organisation of paragraphs require clarifications and improvements. Please answer the following questions: What is overflow: How and where do overflows originate? How long do they propagate? Relative thickness, velocity, range of mass fluxes? Why it is so important in global simulations: e.g Impact on the Global Conveyer belt (MOC) What are the main balance of forces in the overflows! Why is the fine resolution needed, what processes should be resolved in the ideal case? What is the problem in overflow simulation by z-coordinate models, show the numbers! , say predicted temperature 3C higher, etc. It would be great to have an illustration of spurious mixing due to advection+EVD. If other coordinates are better, why are z-coordinates used? What observations and criteria have been used to identify "improvement"?

3. Please characterise the region: main parameters which are important for resolution of overflow: Rossby radius, Ekman depth and maximum/mean topography slopes, slope ratio for each resolution on the sill, as the authors have found this factor is most important. Ekman depth could be estimated from the bottom shear stresses: Hekm = Cdˆ0.5*U_bot/f (Thorpe, 1988) Soulsby (1983).

4. Winton 1998 experiment: "To show the effect of this concept, we simulate the descent of a continuous source of cold water down a shelf break in an idealized configuration of NEMO (with no rotation, comparable to that of Winton et al. (1998))." It is not the Winton, 1998 experiment. Winton compared with an EKMAN – type solution, so the dynamics was rotationally important, his solution was 2D on the f-plane. The solution, shown in fig 9 is not relevant to the baseline study. You consider (fig 9) the propagation of dense boundary layer in a barotopic fluid with a very weak density difference in the plume and ambient waters (0.5C over 3000m depth). So, the balance is between gravity force and friction. Also, you cannot claim that the second case (9b) is worse or better! Is this an effect of EVD or as twice as strong shear, seen in the panel 9b? It is not clear, that solution 9a are physically more consistent compared with 9b. If you want comparisons with analytical solutions, I recommend reproducing Shapiro & Hill 1997 analytical solutions for cascading. This is not completely overflows ( entrainment

is weak), but it is a good test, as approved also by laboratory experiments, (Wobus et al, 2009 and Bruciaferri et al, 2018). My recommendation is to remove this paragraph from the paper, as it is not relevant to the study.

5. I am not convinced that using EVD is a single source of increased simulated mixing when increasing the number of vertical levels. The authors state: What other processes that model start to resolve at finer vertical resolution could affect generation of strong shear and mixing, as inertial or internal waves, topographically trapped Rossby waves? Please, look at high frequency variability at the water column, say, using a Hovmöller diagram. Fig 13a,c, shows the presence of small-scale ( and probably high frequency) features. To my mind it shows presence of internal waves of high amplitude.

6. Another possible cause of an enhanced mixing in the fine vertical resolution is a parameterisation of diffusivity set in a weakly stratified conditions. Indeed, in TKE vertical mixing scheme ( and gls scheme in an strongly stratified conditions), the turbulent length scale is set as $l = 0.1 * TKE^{(1/2)}/N$ and vertical diffusivity $AVT \sim TKE/N$, where TKE is a turbulent kinetic energy, defined by the tke equation but larger by some background value, N is a buoyancy frequency, which differs due to resolution. Subcritical Richardson numbers ($Ri < Ricr \sim 1/4$), responsible for generation of small –scale turbulent mixing are also vertical resolution dependant. Let us consider a plume of dense water of constant density ro_plume propagating downslope in unstratified fluid (of density ro_0) with velocity U.. Velocity shear is S=U/dz, the Richardson number at the edge of the plume is

$Ri = N^2/S^2 = g(Ro\_plume - Ro\_0) * dz/(U^2 * ro\_0)$

will be smaller at the finer vertical resolution which results in more mixing entrainment on the top of the dense plume. This could be examined by comparison of statistics of occurrence the negative (EVD effects) and small positive Richardson numbers at the edge of plume simulations with different resolutions. The other possibility is to check this assumption, to evaluate the number of occurrence of AVT exactly fit to EVD parameterisations (10mˆ2/s, convection) and in the smaller range (∼0.001-1mˆ2/s, Kelvin-Helmholtz instability). In the 1/12 resolution, (Fig 8) I see combination of open-ocean convection (EVD) and shear instability turbulence. How do you explain a much larger area of open convection in the figure 8b, identified by 5-year mean very strong mixing from the surface to the bottom? May be at some point the water of other origin penetrates from the surface to the bottom and mixes with propagating plume?

7. Check consistency of bottom topography, specifically in the "worst case " L300.The authors state: "Bottom topography and coastlines are exactly those of the global 1/12 ORCA12 configuration and are not changed in sensitivity experiments, except when grid refinement is used. In this latter case the refined topography is a bi-linear interpolation of that at 1/12 , so the topographic slopes remain unchanged". It is not seen from the figure 8, where bottom topography is different in simulations L46 and L300. Does adjective TVD scheme work similar in the different vertical resolutions?

Minor comments: Abstract: What observations and criteria have been used to identify "improvement"? Contrary to expectations, in the given numerical set-up, the increase of the vertical resolution "It is found that when the local slope of the grid is weaker than the slope of the topography the result is a more diluted vein. Such a grid enhances the dilution of the plume in the ambient fluid and produces its thickening. Although the greater number of levels allows for a better resolution of the ageostrophic Ekman flow in the bottom layer, the final result also depends on how the local grid slope matches the topography" It is known result from Winton et al. (1998), that the model should resolve slopes and Ekman layer, so if slopes are not resolved, vertical resolution cannot help.

1. From introduction it is not clear, what is overflow, how it is formed and what processes dominates in the dynamics. Even for pure numerical –oriented paper it is important to understand, what should be in the equations and why this resolution is chosen. "An oceanic overflow is a dense water mass" – is this a water mass (object) or process? "Overflows of important magnitude are" – what do you mean under important magnitude? "is balanced by the intrusion of waters from regions different from

where the overflow waters are formed" – please, rephrase it. "For example, the flux of cold waters formed in the Arctic Ocean and the Nordic Seas that enters the North Atlantic with the Denmark Strait and Faroe Bank Channel overflows is balanced by warm and salty Atlantic waters that flows over the Iceland-Scotland Ridge towards the Arctic Ocean via upper ocean currents" It sounds as Atlantic Warm currents are caused by compensation to overflow. "However, the dynamical processes that control overflows have rather small scales" – please, emphasise what small scales processes. What is the main balance in the overflows? Why spurious mixing is considered to be strong? It is not clear from the introduction. Refer to the paper, or just point out what is wrong due to spurious mixing (volume flux, salinity , temperature?) 5. You mention the importance of non-hydrostatic physics and then mention Magaldi & Haine , 2015 paper ( correct reference) showing different contrary results ( see also Wobus et al, 2011). "They also characterized the dependence on various model parameters regarding the mixing of the overflow waters with ambient waters." – I don't understand what do you mean here: mixing depends on parameters? Or something else, which parameters? "found a greater sensitivity of the mixing to horizontal resolution and, but to a lesser extent, to vertical resolution and vertical viscosity" is it resolved horizontal or vertical mixing? Or spurious? How mixing have been examined? "but not the small-scale diapycnal mixing which still needs to be fully parameterized by the turbulent closure scheme." – please, rephrase it. It is true of course, as to resolve diapycnal mixing you need scales up to the dissipative one, which is of 1mm. " a resisting bias in this type of model simulations are likely to contribute." – what is resisting bias? Page 11: "The detailed list of theses experiments" Page 19: (15) "In the case of DSO60.L150 (Fig. 13a,13c) the EVD driven mixing remains confined to a very thin bottom layer below the 15 27.85 isopycnal and very little mixing occurs in the core of the overflow plume" - If you look at the magnitude of near bottom mixing, it is too small to be EVD, probably is it shear-driven Ekman layer; " Intermittent static instabilities occur between the 27.85 and the 27.8 isopycnals, the associated mixing being small since the temperature and salinity gradients are quite small there" - Figure shows very strong mixing>1m^2/s of

high frequency and small scale. As T,S differences are small, Ri numbers to be small there, resulting in a strong intermittent mixing. What frequency and scales are? Is it small positive Ri (Kelvin-Helmholtz instability), or negatve Ri (convection, EVD)?

Thorpe, S.A. (1988) Benthic boundary layers on slopes. In Small Scale Turbulence and mixing in the ocean, Elsevier Oceanography Series, editors Nihoul, J.C.J, Jamart B.M. Soulsby, R.L. (1983). The bottom boundary layer of the shelf sea, in Physical Oceanography of Coastal and Shelf Seas, edited by : B. Johns, Elsevier, New York, USA, 189-266. Shapiro, G. I., & Hill, A. E. (1997). Dynamics of dense water cascades at the shelf edge. Journal of Physical Oceanography, 27(11), 2381-2394. https://doi.org/10.1175/1520-0485(1997)027<2381:DODWCA>2.0.CO;2. Bruciaferri, D., Shapiro, G.I. & Wobus, F. (2018) A multi-envelope vertical coordinate system for numerical ocean modelling. Ocean Dynamics 68(10): 1239-1258.

---

## Author Comment (AC1) · 16 Mar 2020

**Geophysical Model Development Discussion gmd-2019-272-RC1**

**Representation of the Denmark Strait Overflow in a z-coordinate eddying configuration of the NEMO (v3.6) ocean model: Resolution and parameter impacts" by Pedro Colombo et al.**

**Response to the Reviewer 1**

We greatly appreciate comments which helped to largely improve the clarity of our manuscript. In the following, we provide our responses in a point-by-point manner. In our responses below, we use the following legend:

- *Italic characters* for the Reviewers' comments.
- Blue color for our answers to the comments.
- *Blue color in italic* for the revised text, the specific changes being sometimes outlined in *magenta*.

*Introduction*

*Reviewer's comment.*
*2-7 High salinity shelf water which is a source for Antarctic Bottom Water is an overflow too and could/should be mentioned here. Around Antarctica most models struggle to get the dense water from the shelf into the abyssal ocean without entraining too much surrounding water.*
We agree, and we include explicitly this important process in the revised paper (Page 2, starting line 6). Note that this paragraph has also been modified to respond to the comments of reviewer 3.
*"Overflows of important magnitude (not mentioning those crossing deep ocean ridges) are those associated with; the Denmark Strait and the Faroe Bank Channel where dense waters formed in the Arctic and the Nordic Seas flows into the North Atlantic (Girton and Standford, 2003, Brearley et al., 2012, Hansen and Østerhus, 2007); the strait of Gibraltar where dense and saline waters generated in the Mediterranean Sea overflow into the Atlantic Ocean (Baringer and Price, 1997); the strait of Bab-el-Manded where the highly saline Red Sea waters flow into the Gulf of Aden and the Indian ocean (Peters et al., 2005), and the continental shelves of the polar oceans (Killworth, 1977, Baines and Condie, 1998), in particular around Antarctica where the high salinity shelf waters formed in Polynyas ventilate the Antarctic Bottom waters (Mathiot et al., , Purkey et al., 2018). More reference papers can be found in Legg et al. (2009), Magaldi et al. (2015), Mastropole et al. (2017)."*

We added two references.
*Mathiot, P., Jourdain, N.C., Barnier, B., Gallée, H., Molines, J.-M., Le Sommer, J., and Penduff, T., 2012: Sensitivity of coastal polynyas and high-salinity shelf water production in the Ross Sea, Antarctica, to the atmospheric forcing. Ocean Dynamics 62, 701–723 (2012). https://doi.org/10.1007/s10236-012-0531-y.*

*Purkey S.G., Smethie W. M. Jr., Gebbie, G., Gordon, A. L., Sonnerup, R. E., Warner M. J., and Bullister, J. L., 2018: A Synoptic View of the Ventilation and Circulation of Antarctic Bottom Water from Chlorofluorocarbons and Natural Tracers. Annu. Rev. Mar. Sci., 10:8.1–8.25. https://doi.org/10.1146/annurev-marine-121916-063414.*

*Methods*

*Reviewer's comment.*
*Figure 1. As far as I can tell, only section 29, 24, 20, 16 and Denmark Strait have been used. I do not see much value showing all the other sections. I suggest reducing them to the once which are being shown. I am aware that they are meant to show DSOW core.*
Yes, the other sections are used in the study to calculate the path of the overflow. The integral calculations described in Appendix B are performed over the extent of these sections, integration across the section leading to the red spots which identify the path of the DSO in the control simulation.

We modified Fig. 1 which now includes only the 4 most relevant sections (see below).

[Figure]

*Figure 1. Regional model domain. In color the ocean depth. The 250, 500, 1000, 1500 and 2000 meter depth isobaths are contoured in black. The grey box indicates the region where the 2-way grid refinement (1/36° and 1/60°) is applied in some simulations. The location of the various sections used to monitor the model solution are shown by the red lines, and the numbered black lines for the most relevant ones. Section 1 is the reference section chosen for the sill. The thick dotted points indicate the center of the vein of the DSO in the Control simulation as calculated with the formulas given in Appendix B. The green dot indicates the reference point at the sill in order to calculate distances in Fig. 17.*

The other sections are shown in the Appendix B (Fig. B1) where the calculation of the path of the overflow is discussed. The path of the overflow is also shown for the Control and the 1/60°-150 Levels simulations in Fig. B1 (see below).

Figure and text in the Appendix B:

[Figure]

*"Figure B1. Overflow path. Contours show the 500, 1000 and 2000 meter depth isobaths. The location of the various sections used to monitor the model solution are indicated by grey and purple lines. The blue/green dots indicate for each section the location of the center of the vein of the DSO in the Control simulation (blue, DSO12.L46) and in the 1/12° 300 levels simulation (green, DSO12.L300), the blue/green lines outlining the path of the overflow in these simulations."*

The text below has been added in the Appendix B (Page 30).
*"The position of the center of the overflow has been calculated with equations B1 and B2 at each of the 29 sections shown in Figure B1, thus defining the mean path of the overflow in the simulations. This path is used to produce the results shown in Fig. 9 and in Fig.17."*

*Reviewer's comment.*
*Please see my comment how alternatively the DSOW could be tracked, which would not require individual sections.*
Regarding the suggestion of an alternative way to track the DSOW with the minimum bottom temperature, it should work to define the path, but it may also face limitations especially in case of large salinity biases.

Because our sensitivity tests are scanning a large range of parameters, we cannot exclude cases where the bottom temperature signature of the overflow may hardly be different (or even warmer) from that of the ambient fluid, in case for example, of entrainment of highly saline waters. We expect difficulties with such method in simulations where the DSO is considerably unrealistic, which may happen when scanning a large set of parameters and resolutions. Our method based on the calculation of the center of mass and speed of the vein of fluid (Appendix B), which uses potential density and velocity, has the advantage to account for possible compensation in T/S biases and to provide, in addition to the location of the path, the depth of the core (not necessarily at the bottom) from which we can also approximate the thickness of the plume.

We decided to keep our method to calculate the path of the overflow (although we do not use the depth of the plume in the paper).

*Reviewer's comment:*
*Figure 2. It is hard to compare those fields. I would suggest showing the mean from the global configuration and anomalies to the regional setup. In this case it becomes clearer where the differences are. Since both models use the same grid calculating anomalies should be easy.*
We followed this recommendation and plotted the difference in current speed between global and control in subplots 2(c,d) instead of the current speed of Control, but the vectors are the currents of the Control. The vector field in these subplots is still the one from Control. We modified the figure legend and the text of the paper accordingly. At the moment the figures are built from the various subplots by Latex. We shall reduce spaces between subplots, as suggested in the next comment, in the final version of the paper.

The new figure legend is as follows:
*"Figure 2. Surface (a) and bottom (b) mean currents (year 76) in the global ORCA12 simulation. Vectors/Colors indicate current direction/speed in m.s$^{-1}$. Surface (c) and bottom (d) mean currents (year 76) in the regional DSO12.L46 regional simulation. Vectors indicate direction and amplitude of the current. Colors indicate the current speed difference between the global and the regional simulation (in m.s$^{-1}$). Blue/red indicate that the current speed is greater/smaller in the Control (regional) simulation. Vectors at the bottom circulation are scaled by a factor of 7 compared to the surface for visibility reasons."*

Change in the text (Page 8, starting line 2).
*"The large-scale circulation patterns is found to be very similar in both simulations, as illustrated with the surface and bottom currents shown in Fig. 2. The predominant currents such as the East Greenland Current (EGC), the Irminger Current (IC) and the DSO itself are very similar between the global and the regional model. This circulation scheme also compares well with that described from observations in Daniault et al. (2016) and from an ORCA12 model circulation simulation in Marzocchi et al. (2015)."*

*Reviewer's comment:*
*All the subsequent figures have a lot of white spaces between the subplots. If there is any chance to move subplot labels into the figures that would allow to reduce the white spaces and improve the visibility/readability of the figures.*
We agree, and all figures will be modified is a way similar to that applied to Figure 2 before the revised paper is submitted.

*Reviewer's comment:*
*8-14. It appears that the DSOW has a seasonal cycle, which is not present in observations in the Denmark Strait (Jochumsen et al. 2012). Although this is not too critical for this study it shows that likely the formation regions of DSOW in the Nordic Seas are not captured correctly (Våge et al. 2013). That could explain why the transport variability is so low. The seasonal signal usually originates from the EGC and Fram Strait.*
We agree that the seasonal cycle is not realistic and we now mention this in the revised paper (see below). What is important in this figure is that it demonstrates that the regional model is a reliable simulator of what the global model produces in that region, and therefore it is a "good result" that it reproduces this seasonal signal. The reviewer's remark led us to give a greater attention to this signal. Our investigation performed with the regional

model, revealed that it is the barotropic circulation that is driving this seasonal signal (see the new Figure 3). We address this issue by showing and discussing the barotropic transport in Figure 3.

The low values of the transport std shown in Fig. 3 are also a consequence of the sampling used for the model outpouts which are 5-day means (the standard outpout of the global model simulations). Although the regional model outputs are daily means, we used 5-day means in this figure for the purpose of comparison with the global model. When daily means are used the std increases up to 0.7 Sv (more than double), but still remains below what is observed. We do not comment this in the paper, but we indicate in the figure legend that

Modified Figure 3:

[revised manuscript text omitted]

*Figure 5-6 (*Fig. 6-7 in the revised paper*). Is there the chance to include observational values here (CTD casts) along some of these sections? That would help to illustrate how the solution should look like.*

As we say in the paper (section 2.2), the initial conditions of the simulations, which come from a long term (~90 years) global simulation, are significantly different from observations, as the flaws in the representation of the overflows (and other flaws) have modified the mass field (too warm and salty, as discussed). The main objectives of these figures is to compare the solution in twin sensitivity experiments.

To address this comment, we decided to add one figure (Figure 5 in the revised paper, see below), comparing the model solution to observations at a given section. This figure compares the model with observations collected during the ASOF project (Quadfasel, 2004) at the downstream-most section among those shown in the paper (i.e. section 29 in Fig. 1). We chose that section because it is a good illustration of the major flaws of the "end product" in the Control run (the plume is too warm, diluted, does not reach deep enough, and is hardly distinguishable from the ambient fluid). It complements Fig. 4 which shows the "source waters". It also provides guidance regarding assessment of improvements which will be acknowledged if the plume is colder, or deeper, or separated from the ambient fluid by sharper gradients. We modified the text of the paper accordingly. New Figure 5:

[Figure]

*Figure 5: Potential Temperature (°C) at section 29 in (a) the observations (ASOF6-section, Quadfasel, 2004), (b) the 1/60°, 150 levels simulation (annual mean), and (c) the 1/12°, 46 level simulation (annual mean). Red/Green/Black full lines are isopycnals 27.6/27.8/27.85. White lines are isotherms by 1°C interval. For Fig.*

*b), the section 29 is outside (~100 km downstream) the 1/60° AGRIF zoom, so the effective resolution is 1/12°. But the water masses acquired their properties upstream within the 1/60° resolution zoom. Observation data were downloaded at: https://doi.pangaea.de/10.1594/PANGAEA.890362.*

Text changes related to this Fig.5 (page 8, starting line 31):
*"Finally, in order to assess improvements in the sensitivity tests, the major flaws of the control simulation must be described. If similarities with observations are found at the sill, the evolution of the DSO plume in the Irminger basin is shown to be unrealistic in the present setup of the control simulation, and presents the same flaws as in the global run. This is demonstrated by the analysis of the temperature and potential density profiles at the most downstream cross-section (section 29) where the model solution is compared to observations (Fig. 5), and at the other cross-sections along the path of the DSO in the Control simulation (the plots on the left hand side of Fig. 6 and 7). The evolution of the DSO plume as it flows southward along the East Greenland shelf break is represented by a well-marked bottom boundary current (e.g. the bottom currents in Fig. 2) carrying waters of greater density than the ambient waters. Far downstream the sill (section 29) the observations show a well-defined plume of cold water confined below the 27.8 isopycnal under 1500 m depth (Fig. 5a). The bottom temperature is still below 1°C. In the Control simulation (Fig. 5c), one can clearly identify the core of the DSO plume by the 27.85 isopycnal, so it is clear that the plume has been sinking to greater depth as it moved southward. This evolution is only qualitatively consistent with the observations at this section. The modelled plume is significantly warmer and exhibits a core temperature of 3.5° (against 2°C or less in the observations). The plume is also much wider than observed, exhibits much smaller temperature and salinity gradients separating the plume from the interior ocean, indicating a greater dilution with ambient waters. The plume is barely distinguishable from the ambient fluid below 2000m when it is still well marked at that depth in the observations. The sinking and dilution of the plume as it flows southward along the slope of the Greenland shelf is well illustrated in Fig. 6 and 7 (left hand panels) which display the potential temperature at the other sections. If the overflow waters are still well-marked at section 16 (Fig. 6a), it is barely distinguishable from the ambient water at section 29."*

*Maybe just adding density contours would/could already help.*
Main isopycnals (27.6, 27.8 and 27.85) are present in every plot showing vertical sections.

*9-1 It remains unclear where this statement is based on, as far as I can tell observations along these sections are not shown or provided.*
This statement is based on the comparison with the ASOF sections shown in Quadfasel (2004). This remark suggests that this is not clearly formulated. We consider that the addition of the new Figure 5 and the changes in the text to account for it are clarifying this issue.

*Reveiwer's comment:*
*I recommend a re-write of section 8-29 until the results section. The main point is not clear to me. Is it that in the control simulation the temperature in the DSOW layer are more diluted than in the other simulation? If so, this should go in the results section and would also help avoid talking about Figure 7 twice.*
Section 2.3 has three parts that each have a specific purpose to set the paradigm of our study that is: what we shall learn from the regional model will be relevant for the global model, the model solution with the "standard" (i.e. used in most global simulations) parameterization and resolution produces a well-identified overflow so the regional model is relevant for this study, and major flaws in the representation of the overflow properties are identified so it will be possible to assess improvements.

The first part (Page 7 starting line 5) demonstrates that the Regional model reproduces faithfully the global model solution. It ends with: *"Therefore this regional model appears as a reliable simulator of what the global model produces in that region"*. This part is essential part of the paradigm of the study.
The second part (Page 8 starting line 10) describes the properties of the source waters (at the sill) and characterizes their flaws. This part is important because a reasonable degree of realism is needed at the sill for the study of the DSO. This part has been improved by adding tin Fig. 4 the observations by Mastropole et al. (2017). This part ends with the following statement: *"Therefore, despite existing biases, the presence of a well*

*identified dense overflow at the sill confirms the adequacy of the configuration for the sensitivity studies"*. It has been slightly modified to account for the additional plot showing observations.

The third part (Page 8 starting line 29) characterizes, in the control simulation, the flaws in the representation of the overflow along its path, i.e. at the 4 downstream sections for which there are observations from Quadfasel (2004). The main point of this section is to characterize the major flaws of the control experiment, and to demonstrate that they are not different from the flaws of the global model.

The reviewer's comment indicates that its objective of the third part was not made very clear in the text. We consider that the addition of the new Fig. 5 and the changes in the text relative to this part (see our comments about Fig. 5 above) are clarifying this issue especially since we introduce more clearly the objective, this part beginning with (Page 8 starting line 31):
*"Finally, in order to assess improvements in the sensitivity tests, the major flaws of the control simulation must be qualified."*

*Results:*

*Figure 7 (*Fig. 8 in the revised paper*). Is it necessary to show the "warm" >3.6 ◦ C waters? It distracts from the cold DSOW in the Irminger Sea and would allow to get a bit more structure in these plots. Have you tried using anomalies plots here, to make the point clear that with more vertical levels the bottom water gets eroded?*
Interesting comment. We modified the plots using a color palette that emphasizes waters below 4°C. Indeed we found that this change makes the figure more readable.

*Reviewer's comment:*
*Figure 7,10,11 (*8,11,12 in the revised paper*). I think it would help to overlay the DSOW path in these simulations.*
We did not overlay the DSO path on these figures because it tends to mask the details of the overflow properties, especially in their initial descent. We show the overflow path for two of our simulations in Fig. B1 in the Appendix and the reader can refer to this figure. The overflow path, as calculated in Appendix B, is used to make Figures 9.
Fig. 1B and the associated text have been shown above is the discussion of Fig. 1.

*Reviewer's comment:*
*As the authors stated the DSOW is characterized by a temperature minimum, so the path in these simulations could be also defined by the zonal minimum in the regional for each latitude, an alternative way to what the authors use at present.*
Regarding the calculation of the path of the overflow, we already answered this comment before, and we decided to keep our method to calculate the path of the overflow.

*Reviewer's comment:*
*22-14 I am not convinced that reducing the model bias in the source waters will help. Results in Figure 16 (*Fig. 17 in the revised paper*) show that even if modelled temperature would agree with observations, temperatures downstream would end up being warmer than the observations.*
We agree with the referee's analysis of Fig. 16 (Fig. 17 in the revised version). Nevertheless, there is no chance to obtain a realistic representation of the DSO if the source waters (i.e. the waters at the sill and the ambient waters) do not have the correct properties. Therefore, reducing the bias in the sources waters is a necessary condition, but will likely not be sufficient. We slightly modified the text to make this clearer (page 23 line 26):
*"Improved initial and boundary conditions (i.e. correcting for the warm bias of 0.3°C at the sill and for the warm and salty bias of the entrained waters of the Irminger Current) should reduce this difference, but to a point which is difficult to estimate. Either way, the 1.5°C difference shown in Fig. 17 is a quite wide gap that such bias correction will likely not be sufficient to fill."*

*Technical corrections:*
*I could not spot any typos but hope a native speaker might help.*

We did our best with our co-authors.

---

## Author Comment (AC2) · 16 Mar 2020

**Geophysical Model Development Discussion gmd-2019-272-RC2**

**Representation of the Denmark Strait Overflow in a z-coordinate eddying configuration of the NEMO (v3.6) ocean model: Resolution and parameter impacts" by Pedro Colombo et al.**

**Response to the Reviewer 2**

We greatly appreciate comments which helped to largely improve the clarity of our manuscript. In the following, we provide our responses in a point-by-point manner. In our responses below, we use the following legend:

- *Italic characters* for the Reviewers' comments.
- Blue color for our answers to the comments.
- *Blue color in italic* for the revised text, the specific changes being sometimes outlined in *magenta*.

The story, if summarized, is that one should be 'resolving' the topographic slope in the sense that the aspect ratio dz/dx of mesh cells is higher than the slope, and that vertical mesh resolution has to be sufficient to represent the plume (in this manuscript 150 layers provide several points (5-6) across the overflow plume in vertical direction).

*Reviewer's comment 1.*
*My main problem with the manuscript in its current form is that this story is presented as something unexpected and not known. This starts from the abstract and is repeated several times in the text. However, at least as concern the dz/dx ratio, the limitation on this ratio is well known (and authors themselves mention several papers). The second aspect is also general enough to be surprising, of course, the overflow plume has to be resolved vertically, there is no hope on representing the overflow otherwise. The statements like "Contrary to expectations ..." are strange in this context, it is, in contrast, in agreement with expectations. The value of the manuscript is not in the fact that it finds something new and unexpected ("It is found that when the local slope of the grid is weaker than the slope of the topography the result is a more diluted vein" - Is not this known?), but in exploring and documenting precise limitations for the particular ocean circulation model, which will be appreciated by the NEMO community and very likely by other ocean modeling groups. I would recommend that the authors look critically at their statements and adjust the manuscript accordingly (the Abstract, introduction, conclusions in the first turn). I do not think the present form is acceptable.*

We agree that based on the paradigm of convective entrainment expressed by Winton et al. (1998) in their figure 7, we could have expected the sensitivity that we observed. However, we were somewhat surprised by these results because we are working in a range of resolutions that correspond to those for which previous studies (e.g. Winton, 1998) suggest that the representation of the frictional sinking would be achieve with reasonable accuracy (Winton et al. (1998) state in the conclusion of their paper that: "*These conditions imply that resolution on the order of 30–50 m in the vertical and 3–5 km in the horizontal will be needed to represent frictional sinking with reasonable accuracy. This resolution is prohibitive for climate simulations*". With resolution of 5 km to 1 km (i.e. 1/12° to 1/60°) and a large number of vertical levels (150 to 300 levels of resolution of 30 m to 10 m in the depth range 600-1000 m, see Figure A1), we thought possible a behavior that would be dominated by the resolved frictional dynamics.

But finally, our study shows that the convective entrainment paradigm, driven by the EVD parameterization, remains dominant in setting the bottom temperature of the plume. Consequently, we agree to revise our statements regarding our "surprising" or "unexpected" results.

The changes made in the revised paper are listed below:

In the abstract (Page 1, lines 4-5): The text in magenta has been removed.
*" In the given numerical set-up, the increase of the vertical resolution did not bring improvement at eddy-permitting resolution (1/12°)."*

In the Results (Page 14, lines 4-6): The text in magenta has been removed.

*"Finally, the representation of the DSO is even more degraded in the 300 level case, this resolution exhibiting the greatest dilution of the DSO waters among all resolutions. "*

In the Conclusion (Page 24, lines 4-5): The text in magenta has been removed/replaced from the original text.
*" A first result is that the representation of the overflow showed very little sensitivity to any parameter except the horizontal and vertical resolutions. A second result is that,  in the given numerical set-up, the increase of the vertical resolution did not bring any improvement when an eddy-permitting horizontal grid resolution of 1/12° (i.e. ~5km) is used."*

*Reviewer's comment 2.*
*Even in higher resolution runs the bottom topography was kept from 1/12 degree case, and question arises as what will happen if the topography were adjusted according to the resolution. I would appreciate some discussion of the aspect of resolving the topography.*

When horizontal resolution is increased, the bottom topography is bi-linearly interpolated from the 1/12° grid onto the finer grid (1/36° or 1/60°). Therefore, topographic changes still occur at the scale of the finer grid, but the topographic slope remaining constant over a 1/12° blocks (because of the bi-linear interpolation). This is illustrated in Figure 13a,b for example where the original 1/12° (46 levels) and the interpolated on 150 levels topographies can be compared.

*For example, what would happen if 1/12 degree simulations were run on a smoother topography? This might add some useful insight.*

It is very difficult to answer the question without running new model simulations, especially when the bottom topography is realistic and partial steps are used. The study of Penduff et al. (2001) addressed this issue of topographic smoothing and concluded that in an absence of a correct parameterization of current-topography interactions, a certain amount of topographic smoothing have a beneficial impact on geopotential coordinate model solution. Based on these results, we suspect that using an un-smoothed topography in the higher resolution experiments would tend to degrade the results. However, the study of Penduff at al. (2001), focused on the large scale circulation of the South Atlantic (i.e. the Confluence of the Malvinas and Brazil currents, the Zapiola Anticyclone in the Argentinian Basin) did not look at overflows, and we are not confident enough on the generalization of their results to make any comments on that issue in the paper.

We rather not discuss this complex issue in the revised paper.

Penduff, Barnier, Kerbiriou and Verron, 2001: How topographic smoothing contributes to differences between the eddy flows simulated by sigma- and geopotential-coordinate models. J. Phys. Oceanogr., 32, 122-137.

*Reviewer's comment 3.*
*The manuscript is well written, however it tends to overdefine and at too many places phrases could be more concise. Some editing would be good at this level, but it is up to authors.*

We somewhat agree with this comment, and this is likely the reason why the paper is so long. When submitting our paper to GMD, we attempted to make it interesting to and understood by oceanographers, but also by scientists from different scientific fields, as they could bring different and original views to our problems and methods. For this reason, we may have over-defined the context, and few other modelling or methodological aspects of the study, in order to make the paper accessible to scientists of different fields.

We have been through the paper again and attempted to be more concise in our comment, but still keeping our objective of being understood by non-oceanographers.

Some small issues (not all)

Page 2 line 7 check citation style: Corrected
23 'at that resolution' – which one? Can be removed. Removed

Page 3 line 4 'yield to'???? The entire sentence can be written as:

The first complication arises from the the neglect of vertical acceleration in the hydrostatic approximation leading to misrepresentation ... (see 3 above). Corrected

line 30 remove , after (2009) Corrected

page 4 lines 8 and 12 'Despite' and then again 'despite' Corrected
29 'is presented in' – contains "is presented in" is widely used. No change.

page 5 line 24 citation style Corrected

page 6 line 4 citation style Corrected
Caption to Fig.2 an –> and; Surface (a,b) and bottom (c, d) current speed (year 75) in the global ORCA12 (a,c) and regional DSO12.L46 simulations. Only every fourth point is shown.… Corrected in the new legend, since the Figure has been slightly modified to answer comments of Reviewer 1.

page 11 lines 4,5 Following the convention for DSO12.L46, the simulations … Corrected

page 14 line 9 Is NEMO different from all others?
Although we know the general principle of other models (e.g. MIT, HYCOM, FESOM, ROMS), we do not know precisely enough the details of the implementation of their numerics and parameterizations to make pertinent comments of that issue. In the current NEMO framework, the option widely used is to treat the static instabilities with EVD. No change in the text.
line 24 your formula does not express the ratio. Corrected
line 28 250 km wide. Corrected
line 29 when? At time t=0 of the simulation. This is the general definition of initial conditions: the state of the fluid at the beginning of the simulation. To make sure that this is clear, the initial condition is described in one single sentence (page 16, lines3-5).
*"Initial conditions are as follows: a blob of cold water is placed on the bottom of the shelf with a temperature of 10°C, the temperature of the ambient fluid in the rest of the domain being 15°C and the salinity being constant (35 g/kg) in the whole domain."*

line 1 over-resolving the slope vertically worsens the overflow representation Corrected
line 2 there exists or there is Corrected
line 7 Which rationale is meant?
We refer to the rationale of the paper, i.e. what is needed to improve or understanding of the sensitivity of the representation of the DSO in NEMO to the model parameters and resolution ….
But it is absolutely not necessary to recall the main paradigm of the study here. The text now is (Page 18, line 23):
*"Continuing with our rationale, We now evaluate the representation …"*

page 20 line 13 acceleration? or speed-up (units are of velocity) Corrected, speed-up
line 14 5 - 6 points Corrected

---

## Author Comment (AC3) · 24 Mar 2020

**Geophysical Model Development Discussion gmd-2019-272-RC3**

**Representation of the Denmark Strait Overflow in a z-coordinate eddying configuration of the NEMO (v3.6) ocean model: Resolution and parameter impacts" by Pedro Colombo et al.**

**Response to the Reviewer 3**

We greatly appreciated this extensive and detailed review which raised interesting issues and helped to largely improve the clarity of our manuscript. In the following, we provide our responses in a point-by-point manner. In our responses below, we use the following legend:

- *Italic characters* for the Reviewers' comments.
- Blue color for our answers to the comments.
- *Blue color in italic* for the revised text, changes being sometimes outlined in *magenta*.

*Reviewer's comment 1.*
*Absence of figures with observations. It is very hard to follow the text, when the authors refer to figures in the other papers. I found only one plot (Fig 16) to be very informative. Is it possible to plot similar figures from observations? I believe that most of the observed data are present in databases such as EN4.*
We agree with the reviewer that referring to figures published in other papers does not make the reading easy. Following the recommendations, we added figures with observations.

Modification of Figure 4:
This figure now includes the section from Mastropole et al. (M2017) in the paper, such that our assessment of the properties of the overflow "source waters" that compares the model data with M2017 data (Section 2.3) does not require going back and forth between our figure and the figure shown in M2017. We obtained the observation data from R. Pickart group at WHOI. The Figure legend and the text have been modified as follows in the revised version of the paper.

New Figure 4:

[Figure]

(a) Observations    (b) DSO12.L46    (c) DSO12.L46

*"Figure 4. Mean flow characteristics (annual mean of year 76) in the global simulation at the sill. Temperature (°C) in colours and white contours for (a) the observations (Mastropole et al., 2017) and (b) the control simulation (1/12° and 46 vertical levels). Potential density values ($\sigma_0$) are shown by the contour lines coloured in red (27:6), green (27:8) and black (27:85). (c) The velocity normal to the section in the control simulation (southward velocity in blue colour being negative). White lines indicate the 0 ms$^{-1}$ contour (dotted line), the -0.1 ms$^{-1}$ (full line) and the -0.2 ms$^{-1}$ contour (dashed line). The model section being taken along the model coordinate, the topography is slightly different in the model."*

The text now reads (page 8 starting line 24):
*"Fig. 4 presents the characteristics of the mean flow across the sill. The model simulation is compared to the data of Mastropole et al. (2017) who processed over 110 shipboard hydrographic sections across Denmark Strait (representing over 1000 temperature and salinity profiles) to estimate the mean conditions of the flow at the sill (Fig. 4a). The model simulation (Fig. 4b) shows a similar distribution of the isopycnals, specially the location of the 27.8 isopycnal. However, the observations exhibit waters denser than 28.0 in the deepest part of the sill which the model does not reproduce. Large flaws are noticed regarding the temperature of the deepest waters which are barely below 1°C when observations clearly show temperatures below 0°C (also*

*seen in the observations presented in e.g. Jochumsen et al., 2012, Jochumsen et al., 2015, Zhurbas et al., 2016). A bias toward greater salinity values (not shown) is also found in the control experiment which shows bottom salinity of 34.91 compared to 34.9 in the observations shown in Mastropole et al. (2017), but the resulting stratification in density (Fig. 4b) shows patterns that are consistent with observations. The distribution of velocities (Fig. 4c) is also found realistic when compared with observations (i.e. the Fig. 2b of Jochumsen et al., 2012) with a bottom intensified flow of dense waters (up to 0.4 ms⁻¹) in the deepest part of the sill. Although the present setup is designed to investigate model sensitivity in twin experiments and not for comparison with observations ends, the control run appears to provide a flow of dense waters at the sill that is stable over the 5 year period of integration and reproduces qualitatively the major patterns of the overflow "source waters" seen in the observations. Therefore, despite existing biases, the presence of a well identified dense overflow at the sill confirms the adequacy of the configuration for the sensitivity studies."*

Additional Figure (New Figure 5):
We added a new figure comparing the model with observations at the downstream-most section among those shown in the paper (i.e. section 29 in Fig. 1). We chose that section because :

- it is a good illustration of the major flaws of the "end product" in the Control run (the plume is too warm, diluted, does not reach deep enough, and is hardly distinguishale from the ambient fluid), and therefore it complements Fig 4 which shows the "source waters".
- It provides guidance regarding assessment of improvement: improvements will be akcnowledged if the plume is colder, or deeper, or separated from the ambient fluid by sharper gradients.

[Figure]

*Figure 5: Potential Temperature (°C) at section 29 in (a) the observations (ASOF6-section, Quadfasel, 2004), (b) the 1/60°, 150 levels simulation, and (c) the 1/12°, 46 level simulation. Red/Green/Black full lines are isopycnals 27.6/27.8/27.85. White lines are isotherms by 1°C interval. For panel b), the section 29 is outside (~100 km downstream) the 1/60° AGRIF zoom, so the effective resolution is 1/12°. But the water masses acquired their properties upstream within the 1/60° resolution zoom. Observation data were downloaded at* https://doi.pangaea.de/10.1594/PANGAEA.890362.

Modifications brought to the text (page 9, line 6 and following):
*"Finally, in order to assess improvements in the sensitivity tests, the major flaws of the control simulation must be described. If similarities with observations are found at the sill, the evolution of the DSO plume in the Irminger basin is shown to be unrealistic in the present setup of the control simulation, and presents the same flaws as in the global run. This is demonstrated by the analysis of the temperature and potential density profiles at the most downstream cross-section (section 29) where the model solution is compared to observations (Fig. 5), and at the other cross-sections along the path of the DSO in the Control simulation (the plots on the left hand side of Fig. 6 and 7). The evolution of the DSO plume as it flows southward along the East Greenland shelf break is represented by a well-marked bottom boundary current (e.g. the bottom currents in Fig. 2) carrying waters of greater density than the ambient waters. Far downstream the sill (section 29) the observations show a well-defined plume of cold water confined below the 27.8 isopycnal under 1500 m depth (Fig. 5a). The bottom temperature is still below 1°C. In the Control simulation (Fig. 5c), one can clearly identify the core of the DSO plume with the 27.85 isopycnal below 1500 m, so it is clear that the plume has been sinking to greater depth as it moved southward. This evolution is only qualitatively consistent with the observations at this section because the modelled plume is significantly warmer, exhibiting a temperature of 3.5° (against 2°C or less in the observations). The temperature and salinity gradients separating the plume from the interior ocean are smaller than observed, indicating a greater dilution with ambient waters. The plume is barely distinguishable from the ambient fluid below 2000m when it is still well marked at that depth in the observations. The sinking and dilution of the plume as it flows*

*southward along the slope of the Greenland shelf are also well illustrated in Fig. 6 and 7 (left hand panels) which display the potential temperature at the other sections. If the overflow waters are still well-marked at section 16 (Fig. 6a), they are barely distinguishable from the ambient water at section 29."*

*Reviewer's comment 2.*
*Secondly the introduction of the manuscript is not satisfactory written, the style and organisation of paragraphs require clarifications and improvements. Please answer the following questions:*
*What is an overflow? How and where do overflows originate? How long do they propagate? Relative thickness, velocity, range of mass fluxes? Why it is so important in global simulations: e.g Impact on the Global Conveyer belt (MOC)? What are the main balance of forces in the overflows! Why is the fine resolution needed, what processes should be resolved in the ideal case? What is the problem in overflow simulation by z-coordinate models, show the numbers! say predicted temperature 3C higher, etc.*

The information suggested by the reviewers' questions would likely be necessary in a review paper, or a study having for objective to reach the most realistic simulation of the overflows (i.e. accurately comparing with observations), as done in the studies of e.g. Magaldi et al. (2015), Koszalka et al. (2017), Almansi et al. (2017) or Spall et al. (2019). But the scope of our paper is different. The objective is to explore and document the limitations for the NEMO ocean circulation model to represent the overflow of the Denmark Strait, in a context that is relevant for global model simulations, i.e. with resolution and parameterisations now used in global model simulations. We consider that the introduction of the paper is broad enough to introduce the objective. It is already quite long (3 pages), and most questions raised by the reviewer were already addressed, but with less details than the reviewer suggested. Also, answers to some of the reviewer's questions were given in Section 2.3 when we assessed the solution of the control run and describe the major flaws of this simulation.

Nevertheless, the reviewer's comments indicate that the introduction can be improved. So we carefully went through it again and re-structured and modified several paragraphs in an attempt to account for the questions asked.

The introduction is now structured in eight paragraphs which address the following items:
    What is an overflow.
    Why overflows are important.
    Important processes and their representation in OGCMs.
    State of the art in direct simulations of overflows.
    Status of and issues relevant to global eddying models.
    Rationale of the study:  what is needed to improve understanding.
    Objectives of the study.
    Outlines of the paper.

We indicate below the content of each paragraph, and we emphasize in magenta the text that directly answers the reviewer's questions.

(What is an overflow)

[revised manuscript text omitted]

*Quadfasel, D., Käse, R., 2007. Present-day manifestation of the Nordic Seas overflows. In: Schmittner, A., Chiang, J.C.H., Hemmings, S.R. (Eds.), Ocean Circulation: Mechanisms and Impacts, Geophysical Monograph. American Geophysical Union, Washington. 10.1029/1 3GM07.*

*Spall, M.A., R.S. Pickart, P. Lin, W.v. Appen, D. Mastropole, H. Valdimarsson, T.W. Haine, and M. Almansi, 2019: Frontogenesis and Variability in Denmark Strait and Its Influence on Overflow Water. J. Phys. Oceanogr., 49, 1889–1904, https://doi.org/10.1175/JPO-D-19-0053.1*

*It would be great to have an illustration of spurious mixing due to advection+EVD.*
In our "jargon", "spurious" means "excessive and unphysical". We make this clear in the texte (page 2 line 34).

*"However, models using fixed geopotential levels as vertical coordinate (i.e. z-level models) are known to generate spurious (i.e. excessive and non-physical) diapycnal mixing when moving dense overflow waters downslope."*

Quantifying the "spurious" mixing due to numerical schemes has been done in dedicated idealized simulation (e;g. Illicak et al., Ocean Modelling 45–46 (2012) 37–58), but we do not know how to do this in a realistic and forced model simulation. Therefore, we acknowledge that we are not able to provide such an illustration.

*If other coordinates are better, why are z-coordinates used?*
There is no single coordinate that fulfils all the needs of global OGCM (e.g. we do not know about a global implementation of a σ-coordinate model), all coordinates (e.g. geopotential, terrain following, isopycnal) having advantages and disadvantages. The final choice is always pragmatic. NEMO is an OGCM used by a wide scientific and operational community and it is certainly important, if not necessary, to document the sensitivity of the representation of key processes (like DSO) to model parameters.

No change in the text.

*What observations and criteria have been used to identify "improvement"?*
Except for the observations of Mastropole et al. (2017) displayed in Fig. 4 and ASOF6-section of Quadfasel (2004) displayed in Fig. 5, and the bottom temperature at moorings in Fig. 17, we do not use directly observations for our assessment. However, we do use published observations to assess qualitatively the results of our simulations. Qualitative comparison are made  for Fig. 15 with the microstructure measurements from Paka et al. (2013), for Fig. 6,7  with the hydrographic sections from the ASOF project (Quadfasel, 2004) at sections 16, 20, 24 and 29.

The most used criteria to identify improvements between twins simulations is a colder bottom temperature of the DSO waters, as we explained page14, lines 1).
*"From the large set of diagnostics performed to assess the impact of model changes on the DSO, it was found that the"analysis of the bottom temperature in the Irminger Basin is quite a pertinent way to provide a first assessment of the changes in the properties of the overflow. This diagnostic is consequently used to first compare the different sensitivity simulations, an additional diagnostics are used later for more quantitative assessments of the DSO representation."*

Improvements are also identified if major flaws are reduced. These major flaws, identified on the time-mean properties of the overflow of the control simulation (section 2.3, 14-6 15-1,2), are:  too warm bottom temperature, overflow depth not deep enough, weak temperature gradients between the plume and the ambient fluid (a not well-defined dense water plume).

We added a figure (Figure 5) comparing two experiments with observations at section 29 and provide more details on our assessment criteria in first paragraph of the Results section (Section 3, *Page 14 line 4):*
*"Improvements between sensitivity tests are identified when one or several of the major flaws described in the previous section (section 2.3) are reduced. These flaws are; a too warm bottom temperature; an overflow not deep enough; and weak temperature gradients between the plume and the ambient fluid (a not well-defined dense water plume indicating too much dilution)."*

*3. Please characterise the region: main parameters which are important for resolution of overflow: Rossby radius, Ekman depth and maximum/mean topography slopes, slope ratio for each resolution on the sill, as the authors have found this factor is most important. Ekman depth could be estimated from the bottom shear stresses: Hekm = Cdˆ0.5\*U_bot/f (Thorpe, 1988) Soulsby (1983).*
This information is extensively described in the literature (see Quadfasel and Käse, 2007, for example).  The first baroclinic radius of deformation is of the order of 20 km in the Irminger Sea. But this scale is not the one relevant to the instability of the dense water plume (a few kilometres, as we now mention in the introduction when mentioning the important processes).

Looking at the slope ratio for each resolution at and downstream the sill is difficult to use in a realistic setting since it varies greatly from a grid-point to another. This ratio is useful in the idealized experiment that we discuss in Fig. 10 and is chosen to be 5.

The comment on the Ekman depth led us to add a comment regarding its resolution with the vertical resolutions used. This is done in the appendix A in the discussion of Fig. A1 which compares the various vertical resolutions used in the study.

Text added in Appendix A (page 29, line 15):
*"**Vertical Resolutions used:** The variations of the cell thickness as a function of depth is presented in Fig. A1 for the four different vertical resolutions used. A rough estimate of the bottom Ekman layer is given by $h_E = \kappa U_*/f$ (Cushman-Roisin and Beckers, 2011 ) yields $h_E = $ ~45 m in our present model setting for an overflow speed of 0.5 m/s and $U_*$ being calculated from the quadratic bottom friction of the model. Consequently, in the 600 m to 1500 m depth range that correspond to the initial depth range of the overflow, the bottom Ekman layer will only be partially resolved for model vertical resolution of ~10 to 15 m near the bottom, which according to Fig. A1 will happens only for a model resolution of 150 levels (2 to 3 points) and 300 levels (5 to 6 points)."*

We refer to this appendix in the description of the model configuration (Section 2.2, page 6 line 1):
*"The cell-thickness as a function of depth is shown in Appendix A (Fig. A1) and the resolution of the bottom Ekman layer in the different vertical resolution settings is discussed."*

Reference added:

Cushman Roisin B. and Beckers J. M., 2011: Introduction to Geophysical Fluid Dynamics , Physical and Numerical Aspects.  Academic Press, Chap. 8, 219-246.

4. Winton 1998 experiment: "To show the effect of this concept, we simulate the descent of a continuous source of cold water down a shelf break in an idealized configuration of NEMO (with no rotation, comparable to that of Winton et al. (1998))." It is not the Winton, 1998 experiment. Winton compared with an EKMAN – type solution, so the dynamics was rotationally important, his solution was 2D on the f-plane. The solution, shown in fig 9 (Fig. 10 in revised version) is not relevant to the baseline study. You consider (fig 9) the propagation of dense boundary layer in a barotopic fluid with a very weak density difference in the plume and ambient waters (0.5C over 3000m depth). So, the balance is between gravity force and friction.

We agree that we do not reproduce the experiment of Winton et al. (W1988). We only illustrate the concept exposed by the schematic shown in the Fig. 7 of W1988 which does not imply rotation. We realize that our inappropriate reference to the paper of Winton et al. (1988) is the cause of a misunderstanding regarding the purpose of the idealized simulation used to produce our Fig. 10. Our idealized simulation only aims at illustrating how the hydrostatic model NEMO propagates dense water downward a slope, and how this process depends on the vertical resolution. This process is described in Section 2.1, page 6, lines 13-18.
In the idealized set-up of Fig. 10, the dynamics are dominated by advection (driven by horizontal pressure gradient) and diffusion. The model is hydrostatic, so there is no gravity force. Vertical motions are driven by vertical diffusion, and divergence of the horizontal flow that sets the vertical velocity (through non-divergence). Finally, we point out that the ambient fluid is not barotropic but homogeneous, and the density difference between the plume and the ambient fluid (5°C, not 0.5°C) is not weak.

The fact that the realistic model follows this paradigm is illustrated in Fig. R3.4 for the 1/60° resolution and Fig. R3.5 for the 1/12°,  attached to our response to the review. It shows (at section 12 from the sill) that the front of the plume, defined by the 28.85 isopycnal is progressively sinking to great depth under the effect of a negative Richardson number (i.e. under the effect of the EVD parameterization).

The text is modified as follows, removing reference to W1988 where we think not appropriate and bring confusion:

*Page 16 line 27:*
*"An explanation to this is searched for following the paradigm exposed in Fig. 7 of Winton et al. (1998) which states that the horizontal and vertical resolutions should not be chosen independently: the slope of the grid (Δz/Δx) has to equal the slope of the topography (α) to produce a proper descent of the dense fluid."*

*Page 16 line 32:*
*"To show how NEMO follows this concept, we simulate the descent of a continuous source of cold water down a shelf break in an idealized configuration (with no rotation)." .*

*Page 18, lines 1-4:*
*"In that regime, the overflow simulated in the 300 vertical levels run (i.e. with 5 a local grid slope smaller than the topographic slope) presents warmer bottom waters (Fig. 9a) than in the 60 levels run,  and in agreement with the results obtained with the realistic DSO12 configuration."*

Also, you cannot claim that the second case (9b) is worse or better! Is this an effect of EVD or as twice as strong shear, seen in the panel 9b? It is not clear, that solution 9a are physically more consistent compared with 9b.

We agree. In both simulations, the plume propagates downward, essentially due to the high values of the vertical diffusivity (EVD) resulting from the static instability (due to advection of dense fluid over lighter fluid). The bottom water of the plume is colder in the 50 m resolution (60 levels) than in the 10 m resolution (300 levels). Therefore, we consider that the low-resolution case is better regarding the downslope propagation of the cold bottom temperature. We also consider that the upper part of the plume is more coherent in the 10 m resolution case due to a better resolution of the vertical shear (the *tke* scheme being sensitive to vertical resolution). This is explained in pages 17 and 18 in the paper (we removed the reference to Winton 1988 in this part as our idealized simulations do not address the same problem), a paragraph that we reproduce below emphasizing in magenta color the sentences that address these two points:

*"In the absence of rotation, the pressure force pushes the blob over the shelf break and the EVD mixing scheme propagates the cold water down to the bottom as the blob moves toward deeper waters, generating an overflow plume. After about 5 days, the front of the plume has reached the end of the shelf break and entered the damping zone at the right side of the domain, reaching a quasi-stationary regime. In that regime, the overflow simulated in the 300 vertical levels run (i.e. with a local grid slope smaller than the topographic slope) presents warmer bottom waters (Fig. 10a) than in the 60 levels run in agreement with the results obtained with the realistic DSO12 configuration. Note that the vertical shear is more confined in the high-resolution case, which prevents the upward extent of the TKE induced mixing of the upper part of the overflow that is seen in the low-resolution case. Thus, the plume is more consistent in the high-resolution case but present warmer bottom waters. Note that when using a realistic bottom topography, the topographic slope will present large local variations and that it will be almost impossible to match the two slopes over the whole domain in a z-coordinate context. Therefore, increasing the number of vertical levels will not systematically degrade the overflow representation everywhere."*

If you want comparisons with analytical solutions, I recommend reproducing Shapiro & Hill 1997 analytical solutions for cascading. This is not completely overflows (entrainment is weak), but it is a good test, as approved also by laboratory experiments, (Wobus et al, 2009 and Bruciaferri et al, 2018).

We retain the suggestions for future work testing new parameterization of non-hydrostatic effects.

My recommendation is to remove this paragraph from the paper, as it is not relevant to the study.

Having clarified the purpose and context of the idealized simulation and removed the inappropriate link to W1998, we retain this part because we consider it is a good illustration of our interpretation of the behaviour of the cascading in the realistic configuration.

5. I am not convinced that using EVD is a single source of increased simulated mixing when increasing the number of vertical levels. The authors state: What other processes that model start to resolve at finer vertical resolution could affect generation of strong shear and mixing, as inertial or internal waves, topographically

trapped Rossby waves? Please, look at high frequency variability at the water column, say, using a Hovmöller diagram. Fig 13a,c, shows the presence of small-scale ( and probably high frequency) features. To my mind it shows presence of internal waves of high amplitude.

We do not pretend that EVD is the single cause of increased simulated mixing, and the properties of the overflow waters are certainly influenced by other mixing processes (*TKE* or numerically induced) than EVD. The *TKE* closure scheme is NEMO, like many other similar schemes, is consistent with an instability criterion based on a Richardson number (*Ri*). For this reason, the reviewer is right (in the next comment) when suggesting to look at *Ri*. For weak stratifications and significant shear, *TKE* provides large values of *Kz*, sometimes as large as the 10 m$^2$s$^{-1}$ used in EVD. EVD is just a way to "boost" the TKE values in case of static instability ($N^2 > 0$).

In NEMO, the downslope cascading of dense waters from a bottom cell to a deeper bottom cell, which would be driven by vertical acceleration in a non-hydrostatic model, is made by the EVD vertical mixing. Therefore, the dense waters do not sink and accelerate downward but are mixed. Other processes have an impact on the simulated mixing, but by construction of the model, they are not dominant in the representation of the cascading.

The attribution of the high values of the vertical diffusivity coefficient (*Kz*) shown above the 27.85 in Fig.13a,c (1/60°) and not seen in Fig.13b,d (1/12°) is clearly a removal of static instability by EVD, as demonstrated in Figure R3.1 below, which shows hourly value of *Kz* and *Ri* at section 20 at two different times separated by 17 hours. The large *Kz* values between isopycnal 27.80 and 27.85 at hour 254 (Fig. R3.1a) are associated to negative *Ri*, indication removal of a static instability, thus mixing by EVD. At hour 271 (17 hours later, Fig R3.1b), the stratification is stable and *Kz* does not present anymore large values between those isopycnals.

We analysed this period in details (see Fig. R3.2 and Fig. 3.3 attached to this response), and we found that it correspond to the passage through the section of a bottom intensified cyclonic eddy (a bolus of overflow water). The core of the cyclonic eddy (Fig R3.2a), the tangential flow is off-shore and pushes dense water over lighted water, which generate static instability and turns on EVD. The dense water mixes with lighter water below. In the tail of the cyclonic eddy (Fig. R3.2b), the tangential flow is on-shore and does not generate static instability. This does not happens at 1/12° because the horizontal resolution is not enough to well resolve the boluses.

[Figure]

Figure R3.1: Hourly values of the vertical mixing coefficient Kz and of the Richardson number Ri across section 20 at two different times. Ri values of 0.25 are contoured in yellow. Negative Ri values are in dark red.

Therefore, our interpretation that this intermittent, but intense mixing event between those isopycnal is driven by EVD is correct.

Changes in the revised paper:
We modified Fig. 14a,c by picking two different times (those in the Figure above) when this mixing is present and when it is not, to illustrate its intermittency.
We do mention, without providing detailed explanations, that this feature is not seen in the 1/12° it is because it is driven by the cyclonic boluses not resolved at that resolution (section, page 21 line 16):
*"In the case of DSO60.L150 (Fig. 14a,c) a small but noticeable mixing remains confined to a very thin bottom layer below the 27.85 isopycnal, and very little mixing occurs in the core of the overflow plume. Intermittent static instabilities occur between the 27.85 and the 27.8 isopycnals (shown by the large values of Kz in Fig. 14a). Our analysis (no figure shown) indicates that these instabilities are generated by advection toward the deep ocean of bolus of dense water by a cyclonic bottom intensified eddy. After the eddy passed through the section (Fig. 14c) the stratification is again stable. Such feature are not seen in the 1/12° simulation (Fig. 14b,d) because the horizontal resolution does not resolve properly the mesoscale eddies."*

6. Another possible cause of an enhanced mixing in the fine vertical resolution is a parameterisation of diffusivity set in a weakly stratified condition. Indeed, in TKE vertical mixing scheme (and gls scheme in an strongly stratified conditions), the turbulent length scale is set as l= 0.1* TKE^(1/2)/N and vertical diffusivity AVT~TKE/N, where TKE is a turbulent kinetic energy, defined by the tke equation but larger by some background value, N is a buoyancy frequency, which differs due to resolution. Subcritical Richardson numbers (Ri<Ricr~1/4), responsible for generation of small –scale turbulent mixing are also vertical

resolution dependant. Let us consider a plume of dense water of constant density ro_plume propagating downslope in unstratified fluid (of density ro_0) with velocity U. Velocity shear is S=U/dz, the Richardson number at the edge of the plume is

$Ri=N^2/S^2=g(Ro\_plume - Ro\_0)*dz/(U^2*ro_0)$

will be smaller at the finer vertical resolution which results in more mixing entrainment on the top of the dense plume. This could be examined by comparison of statistics of occurrence the negative (EVD effects) and small positive Richardson numbers at the edge of plume simulations with different resolutions. The other possibility is to check this assumption, to evaluate the number of occurrence of AVT exactly fit to EVD parameterisations (10m^2/s, convection) and in the smaller range (∼0.001-1m^2/s, Kelvin-Helmholtz instability). In the 1/12 resolution, (Fig 8, Fig 9 in the revised paper) I see combination of open-ocean convection (EVD) and shear instability turbulence. How do you explain a much larger area of open convection in the figure 8b, identified by 5-year mean very strong mixing from the surface to the bottom? May be at some point the water of other origin penetrates from the surface to the bottom and mixes with propagating plume?

We acknowledge that we did not count the occurrences of EVD. This must be done on-line during integration and this was not in the I/O part of the code, so we stored hourly mean values (i.e. averaged over 8 time steps) to have an estimate of the high frequency motions.  One single EDV event will produce a $Kz > 1$.

We calculated the Richardson number *Ri* as suggested by the reviewer. As shown in Fig. R3.1 above, *Ri* and *Kz* are very consistent, which demonstrate that the *TKE* closure behaviour is very consistent with the stability criterion based on the Richardson number (which is expected). Note that since *Ri* and *Kz* provide almost the same information we do not show *Ri* in the paper.

We modified Fig. 9 and show the mean summer situation, so the winter mixed layer is not present, which allows to better focus on intermediate depths (see below).
It shows that the large *Kz* values between isopycnals 27.6 and 27.8 are not driven from the surface but are generated locally at mid depth. They are driven by the vertical shear existing between the northward surface current passing through the Denmark Strait (the NIIC) which is very variable in position and intensity, and the southward deep current carrying the overflow waters. We notice that the mixing is greater in the high resolution case. Several studies (e.g. Spall et al 2019) show that the NICC can occupy for short periods (few hours to day) the whole strait blocking the passage of the overflow. Our study does not focuses on this process although it is reproduced in our simulations, but of the descent of the dense waters. So our analysis first focuses on the *Kz* near the bottom (below isopycnal 27.85 or 27.8) and then we discuss the values of *Kz* at intermediate depths (Page 16 Line 19):
*"The vertical diffusivity along the path of the overflow is shown in Fig. 9 for the 46 and the 300 level cases (the definition and method of calculation of the overflow path are given in Appendix B). Compared to the 46 level case, the 300 level case (Fig. 9b) exhibits greater values of the diffusion coefficient near and above the bottom along the path of the overflow. This enhanced mixing affects the overflow plume, which 200 km after the sill does not contains waters denser than 27.85, while such waters are still found 300 km down the sill in the 46 level case. The 300 level case also exhibits large values of diffusion coefficient at intermediate depth (between isopycnal 27.8 and 27.6). They are driven by the vertical shear existing between the northward surface current passing through the Denmark Strait (the NIIC) which is very variable in position and intensity, and the southward deep current carrying the overflow waters (e.g. Spall et al 2019). We notice that the mixing is significantly greater in the high resolution case, which indicate that this process could also contribute to the dilution of the overflow plume. However, it does not seem to affect the thickness of the 27.8 isopycnal."*

[Figure]

*"Figure 9. Summer mean (5th year of simulation) of the vertical diffusivity coefficient along the path of the vein calculated for a) simulation DSO12.L46 and b) simulation DSO12.L300. Potential density values (_0) are shown by the contour lines colored in red (27.6), green (27.8) and black (27.85)."*

7. Check consistency of bottom topography, specifically in the "worst case" L300. The authors state: "Bottom topography and coastlines are exactly those of the global 1/12 ORCA12 configuration and are not changed in sensitivity experiments, except when grid refinement is used. In this latter case the refined topography is a bi-linear interpolation of that at 1/12, so the topographic slopes remain unchanged". It is not seen from the figure 8, where bottom topography is different in simulations L46 and L300. Does adjective TVD scheme work similar in the different vertical resolutions?

Topographies are different because the path of the overflow in DSO12.L46 is different from the path in DSO12.L300 (Figure in appendix B).
The model uses a partial step bottom topography, which means that the thickness of the bottom level is adjusted to the real bottom depth. Therefore, the depth does not change when the vertical resolution changes, as it may be the case when a full cell representation of the bathymetry is used (is that latter case the bathymetry is changed to adapt to the thickness of the bottom cell). When horizontal resolution is increased, the topography is linearly interpolated, so the slope is not changed. The consistency of the topography of all simulations can be checked by looking at the bathymetric contours in figures 8, 11 and 12: they are all identical.

**Minor comments:**
Abstract: What observations and criteria have been used to identify "improvement"?
This is now better explained in the paper and does not need to be explicit in the abstract.

Contrary to expectations, in the given numerical set-up, the increase of the vertical resolution "It is found that when the local slope of the grid is weaker than the slope of the topography the result is a more diluted vein. Such a grid enhances the dilution of the plume in the ambient fluid and produces its thickening. Although the greater number of levels allows for a better resolution of the ageostrophic Ekman flow in the bottom layer, the final result also depends on how the local grid slope matches the topography" It is known

result from Winton et al. (1998), that the model should resolve slopes and Ekman layer, so if slopes are not resolved, vertical resolution cannot help.

We removed "contrary to expectation" as this could have been expected, although surprising that this still holds at 1/60° and 300 levels. This could be specific to the NEMO code.

1. From introduction it is not clear, what is overflow, how it is formed and what processes dominates in the dynamics. Even for pure numerical –oriented paper it is important to understand, what should be in the equations and why this resolution is chosen. "An oceanic overflow is a dense water mass" – is this a water mass (object) or process?

We accounted for these comments when revising the introduction of the paper to respond to the second major comments of the reviewer.

"Overflows of important magnitude are" – what do you mean under important magnitude?

We agree that the use of magnitude was not appropriate. We cite the overflow that are important for the general circulation. The text is now:

*"Overflows of importance because of their contribution to the general circulation are …"*

"is balanced by the intrusion of waters from regions different from where the overflow waters are formed" – please, rephrase it. "For example, the flux of cold waters formed in the Arctic Ocean and the Nordic Seas that enters the North Atlantic with the Denmark Strait and Faroe Bank Channel overflows is balanced by warm and salty Atlantic waters that flows over the Iceland-Scotland Ridge towards the Arctic Ocean via upper ocean currents" It sounds as Atlantic Warm currents are caused by compensation to overflow.

Many model studies have demonstrated that in the Atlantic Ocean, weak overflows result in a weak AMOC (e.g. Willebrand, 2001) which is turn reduced the meridional heat transport associated with the northward flowing warm Atlantic waters. In order to simplify the introduction already rich of information we just mention the contribution of the DSO to the deep circulation of the North Atlantic:

*"For the case of the Denmark Strait overflow (DSO hereafter), it feeds the Deep Western Boundary Current in the North Atlantic, and so contributes to the Atlantic Meridional Overturning Cell and the global thermohaline circulation (Dickson and Brown, 1994, …"*

"However, the dynamical processes that control overflows have rather small scales" – please, emphasise what small scales processes. What is the main balance in the overflows? Why spurious mixing is considered to be strong? It is not clear from the introduction. Refer to the paper, or just point out what is wrong due to spurious mixing (volume flux, salinity , temperature?)

We consider that the important changes made to the introduction respond to this comment, as processes, scales of motions are addressed, and spurious mixing is defined…

5. You mention the importance of non-hydrostatic physics and then mention Magaldi & Haine, 2015 paper (correct reference) showing different contrary results (see also Wobus et al, 2011).

"They also characterized the dependence on various model parameters regarding the mixing of the overflow waters with ambient waters." – I don't understand what do you mean here: mixing depends on parameters? Or something else, which parameters?

Our comments regarding the dependence on various model parameters concern the studies of Magaldi et al., 2011, … , which use the hydrostatic version of the MIR GCM, and in this version, the diapycnal mixing is "parameterized" by a turbulent closure scheme since there is not vertical acceleration  (the vertical momentum equation being reduced to the hydrostatic pressure equation). These schemes have parameters to be tuned.

*"Most recent high-resolution regional modelling studies of the Denmark Strait overflow (Magaldi et al., 2011, 2015, Koszalka et al., 2013, 2017, Almnsi et al., 2017, Spall et al., 2019) or the Faroe Bank Chanel overflow (Riemenschneider and Legg, 2007, Seim et al., 2010) have been using hydrostatic model configurations of the MIT OGCM."*

We remove the reference to Magaldi and Haine (2005) in this sentence, although relevant but confusing since this study is cited just above for the use of hydrostatic models.

"found a greater sensitivity of the mixing to horizontal resolution and, but to a lesser extent, to vertical resolution and vertical viscosity" is it resolved horizontal or vertical mixing? Or spurious? How mixing have been examined?

We modified this to emphasize only the sensitivity of resolution, most relevant for our study:
*"For the case of the Faroe Bank Channel overflow for example, Riemenschneider and Legg (2007) found the greatest sensitivity of the mixing in changes in horizontal resolution. "*

"but not the small-scale diapycnal mixing which still needs to be fully parameterized by the turbulent closure scheme." – please, rephrase it. It is true of course, as to resolve diapycnal mixing you need scales up to the dissipative one, which is of 1mm.
We rephrased it as we modified the introduction.
*"Diapycnal mixing processes (e.g. entrainment of ambient waters into the cascading plume by shear-driven mixing, bottom friction, internal wave breaking) have even smaller scales (a few meters to a 1 mm) and cannot be resolved in present ocean models. Their effects are represented by a vertical turbulence closure scheme, the aim of which is to achieve a physically-based representation of this small-scale turbulence."*

" a resisting bias in this type of model simulations are likely to contribute." – what is resisting bias?
A resisting bias is a bias that is not sensitive to the model parameters and cannot be corrected by parameter optimisation. Correcting the bias will require the development of new parameterisations. In the present case, it is more correct to use the word "persisting bias".

We now use "persisting bias" as bias that we haven't corrected.

Page 11: "The detailed list of theses experiments"
Corrected

Page 19: (15)
"In the case of DSO60.L150 (Fig. 13a,13c) the EVD driven mixing remains confined to a very thin bottom layer below the 15 27.85 isopycnal and very little mixing occurs in the core of the overflow plume" - If you look at the magnitude of near bottom mixing, it is too small to be EVD, probably is it shear-driven Ekman layer;
Yes. As we show in Fig. R3.3 added to this review, the EVD mixing is usually acting at the head of the plume and not inside. So within the plume, the mixing is due to the local shear (TKE), but it can be very large in the from of the plume during the phases when it is sinking.
In the paper, we use, to be consistent with the figure:
*"… small but noticeable mixing …"*

" Intermittent static instabilities occur between the 27.85 and the 27.8 isopycnals, the associated mixing being small since the temperature and salinity gradients are quite small there" - Figure shows very strong mixing>1m^2/s of high frequency and small scale. As T,S differences are small, Ri numbers to be small there, resulting in a strong intermittent mixing. What frequency and scales are? Is it small positive Ri (Kelvin-Helmholtz instability), or negative Ri (convection, EVD)?
We clarified this when discussing Fig. 14a,c.

[Figure]

**Fig. R3.2:** Characteristics of the instantaneous (hourly mean) circulation at Section 12 after the sill in simulation DSO60.L150 (1/60°, 150 levels) at three different times. (a) Situation at hour 254 before the arrival of cyclonic eddy. (b) Situation at hour 261 when the bottom intensified cyclonic eddy is passing through the section. (c) Situation at hour 271 when the tail of the eddy is captured (see also Fig. R3.3). The cyclonic eddy is outlined by the dotted line circle in the panel showing the velocity normal to the section

[Figure]

(a) Section across the head of the cyclonic eddy

(b) Section across the center of the cyclonic eddy

(c) Section across the tail of the cyclonic eddy

**Fig. R3.3: Simulation DS060.L150.** Components of the current velocity at Section 20 Schematic illustrating the passage of a cyclonic eddy. Schematics on the right summarise the organisation of the velocity field.

[Figure]

**Fig. R3.4: Simulation DSO60.L150.** Evolution of the hourly Richardson number, *Ri*, at section 12 over a 18 hours period (a plot every 2 hours). Negative values and values below 0.25 are colored in Red. The 0.25 con-tour is shown in yellow. Isopycnals 27.6, 27.8 and 27.85 are plotted in red, green and black respectively. The black arrows show the position of the front of the overflow plume defined as the deepest location of the 27.85 isopycnal, and the vertical dotted grey line indicate the initial position of the front at hour 263. As the front deepens with time, it is always associated with negative values of *Ri* which indicate that the EVD is turned on, illustrating the sinking of dense waters by the EVD parameterisation.

[Figure]

**Fig. R3.5: Simulation DSO12.L150.** Evolution of the hourly Richardson number, *Ri*, at Section 12 over a 32 hours period (a plot every 2 hours). Negative values and values below 0.25 are colored in Red. The 0.25 contour is shown in yellow. Isopycnals 27.6, 27.8 and 27.85 are plotted in red, green and black respectively. The grey arrow show the position of the front of the overflow plume defined as the deepest location of the 27.85 isopycnal, and the vertical dotted grey line indicate the initial position of the front at hour 103. As the front deepens with time, it is always associated with negative values of *Ri* which indicate that the EVD is turned on, illustrating the sinking of dense waters by the EVD parameterisation.

---

## Author Response (AR1)

Grenoble, 24/03/2020

Reference: gmd-2019-272

Dear Editor,

We have revised our manuscript entitled "Representation of the Denmark Strait Overflow in a z-coordinate eddying configuration of the NEMO (v3.6) ocean model: Resolution and parameter impacts" by Pedro Colombo and co-Authors that we submitted as an Article to GMD.

We carefully considered all remarks of the three reviewers and prepared a response for each of them.

So we are pleased to submit the revised paper and the response to the review.

With our best regards,

Pedro Colombo

Institut des Géosciences de l'Environnement, Grenoble

Institut des Géosciences de l'Environnement Université Grenoble Alpes CS 40700 38058 Grenoble cedex 9

www.ige-grenoble.fr

Unité Mixte de Recherche CNRS / IRD / UGA / G-INP UMR 5001 / UR 252

**Geophysical Model Development Discussion gmd-2019-272-RC1**

**Representation of the Denmark Strait Overflow in a z-coordinate eddying configuration of the NEMO (v3.6) ocean model: Resolution and parameter impacts" by Pedro Colombo et al.**

**Response to the Reviewer 1**

We greatly appreciate comments which helped to largely improve the clarity of our manuscript. In the following, we provide our responses in a point-by-point manner. In our responses below, we use the following legend:

- Italic characters for the Reviewers' comments.

- Blue color for our answers to the comments.
- *Blue color in italic* for the revised text, the specific changes being sometimes outlined in *magenta*.

**Introduction**

**Reviewer's comment.**

2-7 High salinity shelf water which is a source for Antarctic Bottom Water is an overflow too and could/should be mentioned here. Around Antarctica most models struggle to get the dense water from the shelf into the abyssal ocean without entraining too much surrounding water.

We agree, and we include explicitly this important process in the revised paper (Page 2, starting line 10). Note that this paragraph has also been modified to respond to the comments of reviewer 3.

"Overflows of importance because of their contribution to the general circulation are those associated with; the Denmark Strait and the Faroe Bank Channel where dense cold waters formed in the Arctic Ocean and the Nordic Seas flows into the North Atlantic (Girton and Standford (2003), Brearley et al. (2012), Hansen and Østerhus (2007)); the strait of Gibraltar where dense saline waters generated in the Mediterranean Sea overflow into the Atlantic Ocean (Baringer and Price (1997)); the strait of Bab-el-Manded where the highly saline Red Sea waters flow into the Gulf of Aden and the Indian ocean (Peters et al. (2005)), and the continental shelves of the polar oceans (Killworth, 1977, Baines and Condie, 1998), in particular around Antarctica where the high salinity shelf waters formed in Polynyas ventilate the Antarctic Bottom waters (Mathiot et al., , Purkey et al., 2018). More reference papers can be found in Legg et al. (2009), Magaldi et al. (2015), Mastropole et al. (2017)."

**We added two references.**

Mathiot, P., Jourdain, N.C., Barnier, B., Gallée, H., Molines, J.-M., Le Sommer, J., and Penduff, T., 2012: Sensitivity of coastal polynyas and high-salinity shelf water production in the Ross Sea, Antarctica, to the atmospheric forcing. Ocean Dynamics 62, 701–723 (2012). https://doi.org/10.1007/s10236-012-0531-y.

Purkey S.G., Smethie W. M. Jr., Gebbie, G., Gordon, A. L., Sonnerup, R. E., Warner M. J., and Bullister, J. L., 2018: A Synoptic View of the Ventilation and Circulation of Antarctic Bottom Water from Chlorofluorocarbons and Natural Tracers. Annu. Rev. Mar. Sci., 10:8.1–8.25. https://doi.org/10.1146/annurev-marine-121916-063414.

**Methods**

**Reviewer's comment.**

Figure 1. As far as I can tell, only section 29, 24, 20, 16 and Denmark Strait have been used. I do not see much value showing all the other sections. I suggest reducing them to the once which are being shown. I am aware that they are meant to show DSOW core.

Yes, the other sections are used in the study to calculate the path of the overflow. The integral calculations described in Appendix B are performed over the extent of these sections, integration across the section leading to the red spots which identify the path of the DSO in the control simulation.

We modified Fig. 1 which now includes only the 4 most relevant sections (see below).

Figure 1. Regional model domain. In color the ocean depth. The 250, 500, 1000, 1500 and 2000 meter depth isobaths are contoured in black. The grey box indicates the region where the 2-way grid refinement (1/36° and 1/60°) is applied in some simulations. The location of the various sections used to monitor the model solution are shown by the red lines.

The other sections are shown in the Appendix B (Fig. B1) where the calculation of the path of the overflow is discussed. The path of the overflow is also shown for the Control and the 1/60°-150 Levels simulations in Fig. B1 (see below).

Figure and text in the Appendix B:

"Figure B1. Overflow path. Contours show the 500, 1000 and 2000 meter depth isobaths. The location of the various sections used to monitor the model solution are indicated by grey and purple lines. The blue/green dots indicate for each section the location of the center of the vein of the DSO in the Control simulation (blue, DSO12.L46) and in the 1/12° 300 levels simulation (green, DSO12.L300), the blue/green lines outline the path of the overflow in these simulations."

The text below has been added in the Appendix B (Page 30).

"The position of the center of the overflow has been calculated with equations B1 and B2 at each of the 29 sections shown in Figure B1, thus defining the mean path of the overflow in the simulations. This path is used to produce the results shown in Fig. 9 and in Fig.17."

**Reviewer's comment.**

Please see my comment how alternatively the DSOW could be tracked, which would not require individual sections.

Regarding the suggestion of an alternative way to track the DSOW with the minimum bottom temperature, it should work to define the path, but it may also face limitations especially in case of large salinity biases. Because our sensitivity tests are scanning a large range of parameters, we cannot exclude cases where the bottom temperature signature of the overflow may hardly be different (or even warmer) from that of the ambient fluid, in case for example, of entrainment of highly saline waters. We expect difficulties with such method in

simulations where the DSO is considerably unrealistic, which may happen when scanning a large set of parameters and resolutions. Our method based on the calculation of the center of mass and speed of the vein of fluid (Appendix B), which uses potential density and velocity, has the advantage to account for possible compensation in T/S biases and to provide, in addition to the location of the path, the depth of the core (not necessarily at the bottom) from which we can also approximate the thickness of the plume.

We decided to keep our method to calculate the path of the overflow (although we do not use the depth of the plume in the paper).

**Reviewer's comment:**

Figure 2. It is hard to compare those fields. I would suggest showing the mean from the global configuration and anomalies to the regional setup. In this case it becomes clearer where the differences are. Since both models use the same grid calculating anomalies should be easy.

We followed this recommendation and plotted the difference in current speed between global and control in subplots 2(c,d) instead of the current speed of Control, but the vectors are the currents of the Control. The vector field in these subplots is still the one from Control. We modified the figure legend and the text of the paper accordingly. At the moment the figures are built from the various subplots by Latex. We shall reduce spaces between subplots, as suggested in the next comment, in the final version of the paper.

**The new figure legend is as follows:**

"Figure 2. Surface (a) and bottom (b) mean currents (year 76) in the global ORCA12 simulation. Vectors/Colors indicate current direction/speed in m.s-1. Surface (c) and bottom (d) mean currents (year 76) in the regional DSO12.L46 regional simulation. Vectors indicate direction and amplitude of the current. Colors indicate the current speed difference between the global and the regional simulation (in m.s-1). Blue/red indicate that the current speed is greater/smaller in the Control/Regional simulation. Vectors at the bottom circulation are scaled by a factor of 7 compared to the surface for visibility reasons."

**Change in the text (Page 8, starting line 5).**

"The large-scale circulation patterns is found to be very similar in both simulations, as illustrated with the surface and bottom currents shown in Fig. 2. The predominant currents such as the East Greenland Current (EGC), the Irminger Current (IC) and the DSO itself are very similar between the global and the regional model. This circulation scheme also compares well with that described from observations in Daniault et al. (2016) and from an ORCA12 model circulation simulation in Marzocchi et al. (2015)."

**Reviewer's comment:**

All the subsequent figures have a lot of white spaces between the subplots. If there is any chance to move subplot labels into the figures that would allow to reduce the white spaces and improve the visibility/readability of the figures.

We agree, and all figures will be modified is a way similar to that applied to Figure 2 before the revised paper is submitted.

**Reviewer's comment:**

8-14. It appears that the DSOW has a seasonal cycle, which is not present in observations in the Denmark Strait (Jochumsen et al. 2012). Although this is not too critical for this study it shows that likely the formation regions of DSOW in the Nordic Seas are not captured correctly (Våge et al. 2013). That could explain why the transport variability is so low. The seasonal signal usually originates from the EGC and Fram Strait.

We agree that the seasonal cycle is not realistic and we now mention this in the revised paper (see below). What is important in this figure is that it demonstrates that the regional model is a reliable simulator of what the global model produces in that region, and therefore it is a "good result" that it reproduces this seasonal signal. The reviewer's remark led us to give a greater attention to this signal. Our investigation performed with the regional model, revealed that it is the barotropic circulation that is driving this seasonal signal (see the new Figure 3). We address this issue by showing and discussing the barotropic transport in Figure 3.

The low values of the transport std shown in Fig. 3 are also a consequence of the sampling used for the model outpouts which are 5-day means (the standard outpout of the global model simulations). Although the regional model outputs are daily means, we used 5-day means in this figure for the purpose of comparison with the global model. When daily means are used the std increases up to 0.7 Sv (more than double), but still remains below what is observed. We do not comment this in the paper, but we indicate in the figure legend that

Modified Figure 3:

"Figure 3. Time evolution of the volume transport of waters of potential density greater than 27.80 kgm-3 at the sill section (Section 1 in Fig. 1) in the Control (blue line) and the Global (green line) simulations (the latter providing the open boundary conditions). Annual mean and std (in Sv) are indicated for every individual year of simulation. The depth-integrated (barotropic) transport is shown for the Control simulation (purple line). 5-day mean values are used to produce this figure."

**Modified text (page 8, line 19):**

"The standard deviation computed from 5-day outputs (~0.3 Sv in the control run, increasing to 0.7 Sv when calculated from daily values) is rather small when compared to the 1.6 Sv of Macrander et al. (2005). The modelled flow of dense waters presents a marked seasonal cycle which is not present in observations (Jochumsen et al. 2012). This signal is the signature of the large seasonality of the barotropic flow (Fig. 3) that constrains the whole water column."

Figure 4. I would swop (a) and (b) so you can avoid starting in line 8-16 with Figure 4b and later going back to Figure 4a.

This figure has been modified to include a plot showing the observations of Mastropole et al. (2017). The Figure legend and the text have been modified as follows in the revised version of the paper.

---

## Author Response (AR2)

Grenoble, 19/05/2020

[Figure]

Reference: gmd-2019-272

Dear Editor,

We have revised our manuscript entitled "Representation of the Denmark Strait Overflow in a z-coordinate eddying configuration of the NEMO (v3.6) ocean model: Resolution and parameter impacts" by Pedro Colombo and co-Authors that we submitted as an Article to GMD. We carefully considered all your remarks and corrected the paper accordingly.

So we are pleased to submit the revised paper. In this version of the paper, the changes in the text in response to the comments are outlined in blue color to be easily identified. This will be corrected after the final decision. Our detailed response to your comments is attached to this letter.

With our best regards,

Pedro Colombo

Institut des Géosciences de l'Environnement, Grenoble

Institut des Géosciences de
l'Environnement
Université Grenoble Alpes
CS 40700
38058 Grenoble cedex 9

www.ige-grenoble.fr

Unité Mixte de Recherche
CNRS / IRD / UGA / G-INP

[Figure]

[Figure]

[Figure]

[Figure]

**Detailed response to the Editor's comments**

We provide here our responses in a point-by-point manner to the revisions recommended by the Editor, using the following legend:
**- Bold characters to recall the Reviewers' comments.**
*- Italic characters for our answers to the comments.*
- "Blue color in quotes" for the text of the paper, changes being outlined in magenta.

Note that

**\* Please clarify the final sentence of the caption to figure 4 -- are the sections themselves slightly different (different location/angle?) leading to the difference in topography? There seems to be quite a large difference in panel a).**

*Response:*
*The model section was made along the coordinate of the model, which more or less follows a longitude line, when the observations section has a different angle and is an analysis of a large number of observations collected nearby the section but not necessarily exactly along the section (Fig. 1 of Mastropole et al, 2017). This explains the different topography.*
*We changed the figure and we now plot a section that is very close to the one of the observations. The difference in topography is much less, and the agreement with observations is improved. We slightly modified the figure legend and the corresponding text of the paper.*

*Change in the paper:*
*New figure 4 and its legend:*

[Figure]

"Figure 4. Mean flow characteristics (annual mean of year 76) in the global simulation at a section across the sill. Temperature (°C) in colours and white contours for (a) the observations (Mastropole et al. (2017)) and (b) the control simulation (DSO12.L46, 1/12° and 46 vertical levels). Potential density values ($\sigma_0$) are shown by the contour lines coloured in red (27.6), green (27.8) and black (27.85). (c) The velocity normal to the section in the control simulation (southward velocity in blue colour being negative). White lines

indicate the 0 ms$^{-1}$ contour (dashed-dotted line), the -0.1ms$^{-1}$ (full line) and the -0.2ms$^{-1}$ contour (dashed line)."

*Change in the text (page 8): the special comment regarding isopycnal 27.8 is removed because this isopycnal is not anymore special at this location.*
"Compared with the compilation of observations of Mastropole et al. (2017) (Fig. 4a) the model simulation (Fig. 4b) shows a consistent distribution of the isopycnals, specially the location of the 27.8 isopycnal."

**\* Could the cited data in the figure 5 caption be added to the bibliography?**
*Response:*
*We added the link to the data to the reference to the cruise report of Quadfasel (2010) made in the Figure legend.*
*Change in the paper:*
*The legend now is:*
"Figure 5. Potential Temperature (°C) at section 29 in (a) the observations (ASOF6-section, Quadfasel (2010)), (b) the 1/60°, 150 levels simulation (annual mean), and (c) the 1/12°, 46 level simulation (annual mean). Red/Green/Black full lines are isopycnals 27.6/27.8/27.85. White lines are isotherms by 1°C interval. In panel b), the section 29 is outside ( 100 km downstream) the 1/60° AGRIF zoom, so the effective resolution is 1/12°. But the water masses acquired their properties upstream within the 1/60° resolution zoom. "
*And the reference below has been added to the reference list, nad the reference to Quadfasel (2004) was removed (because an older version of the same report):*
"Quadfasel, D., 2010: Summary Cruise Report RV METEOR Cruise M82-1, Reykjavik - St. John's, 3. July - 2. August 2010. University of Hamburg, 9 pp. https://doi.pangaea.de/10.1594/PANGAEA.890362."

**\* Page 24 lines 25-27: Is an improved representation of topography expected, given that the topography is interpolated from the 1/12 degree configuration?**
*Response:*
*Yes, but only is special cases. A thin V-shaped channel can be represented with a U-shape at 1/12° if its width is close that that of the resolution, but will have a V-shape channel at 1/60° even if the topography is interpolated linearly. But such special case is important here, this is why we mentioned it. We attempt to clarify this with a slight modification of the text. The new text reads:*
*Change in the paper:*
"Finally, we would like to point out that the increase in resolution may also improves the representation of topography, despite it is linearly interpolated from the 1/12°. For example, the thin v-shaped channel over the sill (Fig. 4) which may be represented by a U-shape at coarse resolution, may again be represented with a V-shape as resolution increases."

**p2 l5 grammar at "thick of"**
"Overflows are often structured as plumes or boluses of dense fluid **thick of** a few hundred meters"
*Replaced by:*
"Overflows are often structured as plumes or boluses of dense fluid of a few hundred meters thickness**"**

**p2 l32 citation typesetting**
*Corrected.* (Girton and Standford (2003)).

**p3 l24 "Chanel" typo**
*Corrected "Channel"*

**p4 l4 broken citation**
*Corrected* "Chang et al. (2009) "

**p6 l13 citation typesetting**
Corrected, it now reads: "(Madec et al. ( 2016), Reffray et al., 2015))"

**p6 l18 citation typesetting**
*Corrected* "(Reffray et al. (2015))"

**p6 l27 citation typesetting**
*Corrected* " (Brodeau et al. (2010))"

**p8 l5 "patterns is" grammar**
*Corrected* "Patterns are"

**p9 l2 "observation ends" grammar**
*Corrected* "observational ends"

**p16 l30 grammar at "thicken"**
*Corrected* "thickens"

**p20 l3 grammar**
*Corrected, the previous sentence:*
"Once reached a vertical resolution that is adequate for a specific horizontal resolution for a given slope, increasing the vertical resolution will deteriorate the DSO representation by introducing excessive vertical mixing."
*has been replaced by:*
"Once a vertical resolution adequate for a specific horizontal resolution is set for a given slope, an increase in vertical resolution will deteriorate the DSO representation by introducing excessive vertical mixing."

**p24 l12 "somehow summarizes" unusual phrasing**
*Corrected,* "somehow summarizes" *replaced by* "summarizes"

**p24 l30 grammar at "to eddy-permitting"**
*Corrected,* "from eddy-permitting"

**p25 l2 "worst" instead of "worse"?**
*Agreed, corrected.*

**p26 l1 "best performing" instead of "most performing"?**
*Agreed, corrected.*

**p27 l6 "problem" instead of "problematic"**
*Agreed, corrected.*

**p27 l12 grammar**
*Corrected, the previous sentence:*
"This conclusion draws our attention to a limitation for future global simulations in z-coordinate since all flows that are topographically constrained do not flow along the same topographic slope."

*Has been replaced by:*

This conclusion leads us to draw attention to the limitation of z-coordinate global simulations to correctly represent all major overflows since topographic slopes largely vary in the world ocean."

**p27 l15 remove "all in all"?**
*Agreed, removed.*

**p29 l7 "an horizontal"**
*Corrected  at 3 occurences:* "a horizontal"

**Author contributions grammar at "where" (twice)**
*Corrected both occurences* "were"

*Changes to Acknowledgements*
*We also thank our WHOI colleagues who provided the observation data to produce Fig. 4a, and D. Quadfasel who made cruises observations fully available on the web. The acknowledgements now read as follows.*

[revised manuscript text omitted]

---

## Author Response (AR3)

Grenoble, 13/06/2020

Reference: gmd-2019-272

Dear Editor,

We are pleased to know that our manuscript entitled "Representation of the Denmark Strait Overflow in a z-coordinate eddying configuration of the NEMO (v3.6) ocean model: Resolution and parameter impacts" by Pedro Colombo and co-Authors that we submitted as an Article to GMD is accepted for publication subject to technical correction. We carefully considered the corrections recommended and corrected the paper accordingly.

So we are pleased to submit the final paper for the production process.

With our best regards,

Pedro Colombo

Institut des Géosciences de l'Environnement, Grenoble

Institut des Géosciences de l'Environnement
Université Grenoble Alpes
CS 40700
38058 Grenoble cedex 9

www.ige-grenoble.fr

Unité Mixte de Recherche
CNRS / IRD / UGA / G-INP
UMR 5001 / UR 252